# Towards Flexible Visual Relationship Segmentation

**Fangrui Zhu**[1]  **Jianwei Yang**[2]  **Huaizu Jiang**[1]
[1]Northeastern University  [2]Microsoft Research
`https://neu-vi.github.io/FleVRS`

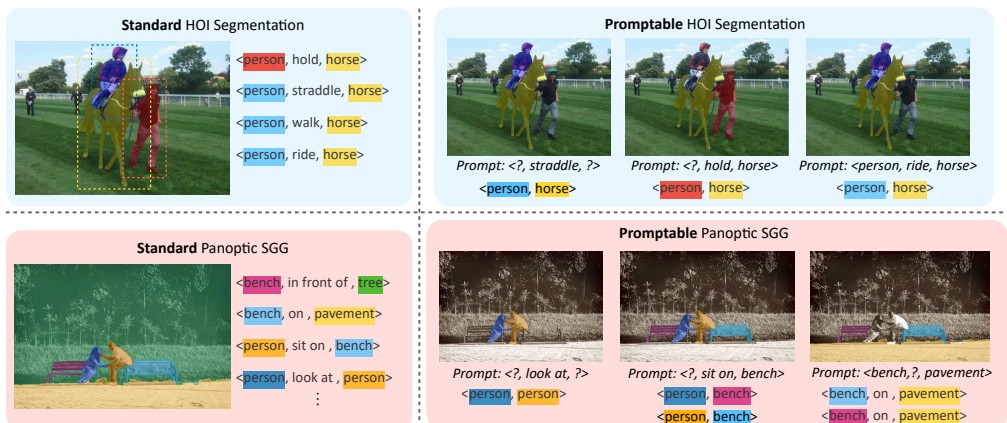

Figure 1: `FleVRS` is a single model trained to support **standard**, **promptable** and **open-vocabulary** fine-grained visual relationship segmentation (`<subject` mask, relationship categories, `object mask>`). It can take images only or images with structured prompts as inputs, and segment all existing relationships or the ones subject to the text prompts.

## Abstract

Visual relationship understanding has been studied separately in human-object interaction (HOI) detection, scene graph generation (SGG), and referring relationships (RR) tasks. Given the complexity and interconnectedness of these tasks, it is crucial to have a flexible framework that can effectively address these tasks in a cohesive manner. In this work, we propose `FleVRS`, a single model that seamlessly integrates the above three aspects in standard and promptable visual relationship segmentation, and further possesses the capability for open-vocabulary segmentation to adapt to novel scenarios. `FleVRS` leverages the synergy between text and image modalities, to ground various types of relationships from images and use textual features from vision-language models to visual conceptual understanding. Empirical validation across various datasets demonstrates that our framework outperforms existing models in standard, promptable, and open-vocabulary tasks, *e.g.*, **+1.9** $mAP$ on HICO-DET, **+11.4** $Acc$ on VRD, **+4.7** $mAP$ on unseen HICO-DET. Our `FleVRS` represents a significant step towards a more intuitive, comprehensive, and scalable understanding of visual relationships.

## 1  Introduction

An image is not merely a collection of objects. Understanding the visual relationships between different entities at pixel-level through segmentation is a fundamental task in computer vision, which has broad applications in autonomous driving [28, 58], behavior analysis [65, 67], navigation [10, 15, 22, 27], *etc*. Furthermore, segmenting relational objects extends beyond mere detection, playing a

38th Conference on Neural Information Processing Systems (NeurIPS 2024).

| Method | Standard | | Promptable | Open-vocabulary | One/Two-stage Model |
| --- | --- | --- | --- | --- | --- |
| | HOI | SGG | | | |
| RLIPv2 [92] | ✓ | ✓ | ✗ | ✓ | Two |
| UniVRD [101] | ✓ | ✓ | ✗ | ✗ | Two |
| SSAS [38] | ✗ | ✗ | ✓ | ✗ | One |
| GEN-VLKT [47] | ✓ | ✗ | ✗ | ✓ | One |
| FleVRS (Ours) | ✓ | ✓ | ✓ | ✓ | One |

Table 1: **Comparisons with previous representative methods in three aspects of model capabilities**. To the best of our knowledge, our `FleVRS` is the first one-stage model capable of performing standard, promptable, and open-vocabulary visual relationship segmentation all at once.

crucial role in improving visual understanding and providing a more comprehensive abstraction on the visual contents and interactions among them.

Ideally, a visual relationship segmentation (VRS) model should demonstrate flexibility across three key dimensions. 1) **Capability of segmenting various types of relationships**, including both human-centric and generic ones. These relationships are defined as triplets in the form of `<subject, predicate, object>`. Human-object interaction (HOI) detection [4, 18], which we adapt into HOI segmentation in our work, exemplifies this capability, such as `<person, ride, horse>` in Fig. 1. Panoptic scene graph generation (SGG) [55, 81, 85], captures generic spatial or semantic relationships among pairs of objects in an image, *e.g.*, `bench on pavement` in Fig. 1. A unified model that can handle these tasks concurrently is essential, as it eliminates the need for separate designs and modifications for each specific task. 2) **Grounding of relational subject-object pairs with different prompts**. Given various textual prompts, the model should output the desired entities and relationships, facilitating a more natural and intuitive user interface. For instance, it should be able to detect just the `person` in an image or all possible interactions between a `person` and a `horse`, as illustrated in Fig. 1. 3) **Open-vocabulary recognition of visual relationships**. In realistic open-world applications, the model should generalize to new scenarios without requiring annotations for new concepts not seen during training. This capability includes detecting novel objects, relationships, and their combinations.

Existing models in visual relationship segmentation (VRS) have targeted aspects of the desired capabilities but fall short of providing a comprehensive solution, as detailed in Tab. 1. Models have typically focused on tasks like human-object interaction (HOI) detection [23, 35, 47, 57, 72, 95, 97, 108] and panoptic SGG [50, 71, 81, 85, 93, 104]. Although models such as [92, 101] have attempted to unify VRS under a single framework, they need additional pretraining on HOI datasets (Tab. 1) and lack features such as promptable segmentation, which allows for dynamic entity and relationship generation based on textual prompts, as well as capabilities for open-vocabulary promptable segmentation. Efforts to detect instances referred to by textual prompts have been made [21, 38, 75, 107], but these models fail to capture all desired entities or relationships comprehensively and struggle with classifying multi-label interactions between the pairs, limiting their effectiveness in complex scenarios. Although recent vision-language grounding models like [29, 52, 83] and multimodal large language models such as [2, 5, 76, 78, 88] exhibit enhanced capabilities in grounding instances specified by free-form text and show strong generalization over novel concepts, they still do not generate the required pairs in the format of segmentation masks. Furthermore, these models require significant computational resources and additional vision models for precise localizations. For open-vocabulary VRS, existing works [47, 91, 92] leverage textual embeddings to transfer knowledge. However, models [91, 92] fall short in grounding diverse prompts, while [47] is exclusively designed for HOI detection, not generic VRS.

To address the limitations in existing models, we introduce `FleVRS`, a flexible one-stage framework capable of performing standard, promptable, and open-vocabulary visual relationship segmentation *simultaneously*. Our approach integrates human-centric (HOI segmentation) and generic VRS (Panoptic SGG) by adopting SAM [36] to unify different types of annotations into segmentation masks and using a query-based Transformer architecture that outputs triplets in the format `<subject, predicate, object>`. The model enhances its interactive capabilities by accepting textual prompts as inputs. These prompts are converted into textual queries that assist the decoder in accurately identifying and localizing objects within the relationships. Additionally, we unify the labels from different datasets into a shared textual space, transforming classification into a process of matching with a set of textual features. Leveraging textual features from the CLIP model [64], we enable the effective matching of visual features with textual knowledge of novel concepts. This design

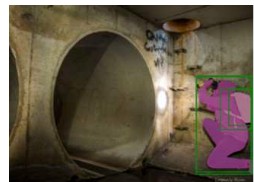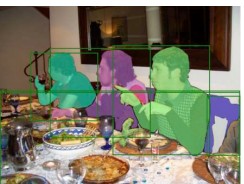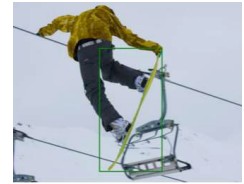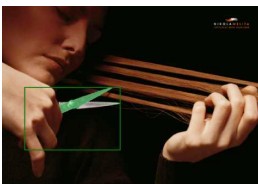

Figure 2: **Examples of converting HOI detection boxes to masks.** We filter out low-quality masks during training by computing IoU between the mask and box.

inherently supports open-vocabulary and promptable relationship segmentation without pre-defining the number of object or predicate categories, facilitating dynamic and extensive adaptability.

Our `FleVRS` proposes a unified framework that integrates standard, promptable, and open-vocabulary VRS tasks into a single system, as detailed in Tab. 1, providing greater flexibility compared to existing methods. It employs a mask-based approach to effectively manage various VRS tasks, enabling adaptation to different types of annotations, including HOI detection and panoptic SGG. Our architecture incorporates dynamic prompt handling, which supports both prompt-based and open-vocabulary settings, allowing our model to combine promptable queries with open-vocabulary capabilities to ground novel relational objects.

We evaluate our `FleVRS` on standard, promptable, and open-vocabulary VRS tasks, *i.e.*, HOI segmentation [4, 18] and panoptic SGG [85]. Crucially, we demonstrate competitive performance from three perspectives – standard (**40.5** *vs.* **39.1** $mAP$ on HICO-DET [4]), promptable (**56.8** *vs.* **33.5** sIoU on VRD [55]), and open-vocabulary (**31.7** *vs.* **25.6** $mAP$ for "unseen object" on HICO-DET [4]) visual relationship segmentation.

In summary, our main contributions are as follows: 1) We introduce a flexible one-stage framework capable of segmenting both human-centric and generic visual relationships across various datasets. 2) We present a promptable visual relationship learning framework that effectively utilizes diverse textual prompts to ground relationships. 3) We demonstrate competitive performance in both standard close-set and open-vocabulary scenarios, showcasing the model's strong generalization capabilities.

## 2  Related Work

**Visual Relationship Detection** (VRD) is split into two lines of works, including human-object interaction (HOI) detection [4, 18] and panoptic scene graph generation (SGG) [37, 85]. They are defined as detecting triplets in the form of `<subject, predicate, object>` triplet, where `subject` or `object` includes object box and category. HOI detection aims to detect human-centric visual relationships, while PSG focuses on generic object pairs' relationships. Previous works [1, 13, 16, 32, 35, 41, 41, 47, 57, 81, 89, 95, 96, 97, 100, 104, 105, 106] usually train specialist models on a single data source and tackle them separately. Departing from this traditional bifurcation, UniVRD [101] initiated the development of a unified model for VRD, with subsequent efforts like [91, 92] advancing relational understanding through large-scale language-image pre-training. Unlike the two-stage approach of [92], which performs object detection before decoding visual relationships, our method employs a one-stage design that decodes objects and their relationships simultaneously. Crucially, our model extends beyond standard VRD capabilities to support promptable and open-vocabulary visual relationship segmentation, enhancing detailed scene comprehension.

**Referring relationship and visual grounding.** The most relevant work to ours is referring visual relationship introduced in [38], where the model detects the subject and object depicting the structured relationship `<subject, predicate, object>`. One-stage [38, 75], two-stage [63, 107] and three-stage [21] methods are proposed to localize the two entities' boxes iteratively based on the given structured prompt `<subject, predicate, object>`. Unlike these methods, our approach allows for more *flexible* textual prompts without requiring the complete specification of the triplet. As shown in Fig. 1, our model can handle queries that include a single item (e.g., `predicate`) or a combination of two (e.g., `predicate` and `object`). Additionally, our method is capable of performing standard and open-vocabulary VRS. Visual grounding represents another related area, where models output bounding boxes [7, 8, 20, 29, 40, 52, 83] or object masks [9, 17, 44, 45, 51, 56, 79, 87, 99] in response to textual inputs. This process requires reasoning over the entities mentioned in the text to identify the corresponding objects in the visual space. However, our task fundamentally differs from this.

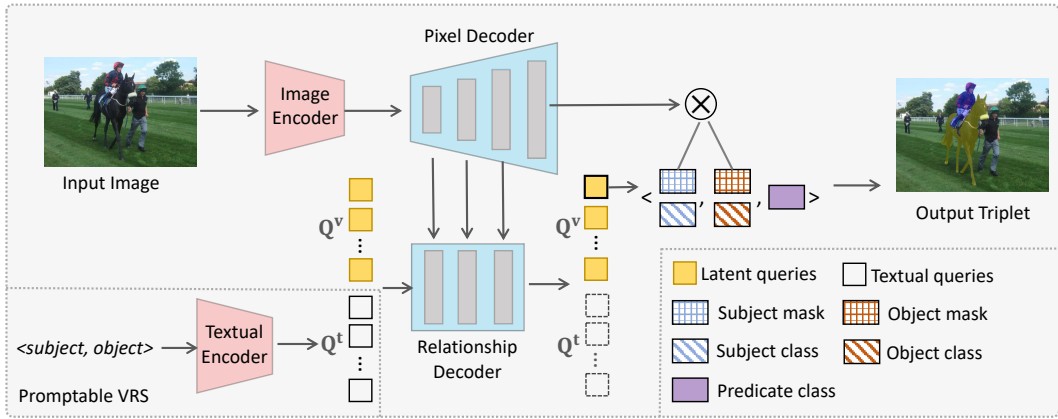

Figure 3: **Overview of** `FleVRS`**.** In standard VRS, without textual queries, the latent queries perform self- and cross-attention within the relationship decoder to output a triplet for each query. For promptable VRS, the decoder additionally incorporates textual queries $\mathbf{Q^t}$, concatenated with $\mathbf{Q^v}$. This setup similarly predicts triplets, each based on $\mathbf{Q^v}$ outputs aligned with features from the optional textual prompt $\mathbf{Q^t}$.

In our `FleVRS`, the promptable VRS task goes beyond mere identification; it involves outputting segmentation masks for both subject and object pairs along with categorizing their relationships. This capability is essential for understanding and interpreting complex relational dynamics.

**Vision and language models** Recent advancements in large-scale pre-trained vision-language models (VLM) [39, 64, 68, 69, 82, 94] and multimodal large language models (MLLM) [2, 5, 76, 78, 88] have demonstrated impressive performance and generalization capabilities across a variety of vision and multimodal tasks [36, 109, 110]. However, these models primarily focus on entity-level generalization, with open-vocabulary VRS receiving less attention. While recent efforts in zero-shot HOI detection [47, 61, 80, 92] often utilize CLIP [64] for category classification, their open-vocabulary capabilities lack the flexibility needed for prompt-driven input. Although current VLMs and MLLMs are adept at grounding novel concepts from text, they require significant computational resources and additional visual models, and cannot directly generate comprehensive segmentation masks for subject and object pairs. In contrast, our `FleVRS` provides a lightweight solution that effectively supports various types of open-vocabulary VRS, enabling category classification and the integration of novel concepts from prompts.

## 3  Method

### 3.1  Overview

**Standard VRS**. Given an image $\mathbf{I}$, the goal of *standard* visual relationship segmentation (VRS) is to detect all the visual relationships of interest, either human-centric (*i.e.*, HOI detection) or the generic ones (SGG), in terms of triplets in the form of `<subject, predicate, object>` (masks and object categories of `subject` and `object`, and the `predicate` category). The `subject` is always `human` in HOI detection, whereas it can be any type of object in SGG (may or may not be `human`). We consider the panoptic setting [85] of SGG, where a model needs to generate a more comprehensive scene graph representation based on panoptic segmentation rather than rigid bounding boxes, providing a clear and precise grounding of objects. To produce fine-grained masks, we convert existing bounding box annotations from HOI detection datasets [4, 18] into segmentation masks using the foundation model SAM [36], as illustrated in Fig. 2. We employ a filtering approach based on Intersection over Union (IoU) to filter out inaccurate masks. Details are in Appendix.

**Promptable VRS.** Our `FleVRS` optionally accepts textual prompts as inputs, enabling users to specify visual relationships for promptable VRS. It accommodates three types of structured text prompts: a single element (e.g., `<?, predicate, ?>`), any two elements (e.g., `<subject, predicate, ?>`, `<subject, ?, object>`), or all three elements. Consequently, the model outputs only the triplets that match the specified elements in the prompt, as depicted in the right column of Fig. 1. Without textual prompts, it functions as a standard VRS model, exhaustively generating all possible triplets, illustrated in the left part of Fig.1.

**Open-vocabulary VRS.** In practice, it's essential for a VRS model to adapt to new concepts, including new categories of entities (*i.e.*, `subject` and `object`), `predicates`, and their various combinations. Expanding these concept vocabularies to encompass a wider range is particularly challenging due to the vast potential combinations and the long-tail distribution of these categories. Thus, our goal is to equip the model to operate in an open-vocabulary setting, where it can effectively handle these diversities. It it important to note that the above three capabilities are complementary; for example, the text prompts in promptable VRS can include novel object or predicate categories.

To this end, we propose integrating the above three aspects into a *single unified* framework. Since these settings are complementary, a general-purpose model should be capable of performing various combinations of these three functions. Additionally, their inherent similarities make it more intuitive to consolidate them within a flexible, unified approach.

## 3.2 Model Architecture

Inspired by the success of Transformer-based segmentation models [6, 109], we design a dual-query system for our VRS model, illustrated in Fig. 3. Latent queries, a set of learned embeddings, generate triplets (which may be empty) to formulate output masks and relationship categories. For promptable VRS, textual queries derived from input prompts are incorporated. We employ an image encoder and a pixel decoder to extract visual features, coupled with a relationship decoder that processes `<subject, object>` pairs and their interrelations. For open-vocabulary VRS, our approach shifts from traditional classification to a matching strategy that aligns visual and textual features for both object and predicate categories, enhancing the model's adaptability to new concepts. Each component of this architecture is elaborated further below.

**Image Encoder.** Specifically, given the image $\mathbf{I} \in \mathbb{R}^{H \times W \times 3}$, it is first fed into the image encoder $\mathbf{Enc_I}$ to obtain multi-scale low-resolution features $\mathbf{F} = \left\{ \mathbf{F}_s \in \mathbb{R}^{C_\mathbf{F} \times \frac{H}{s} \times \frac{W}{s}} \right\}$, where the stride of the feature map $s \in \{4, 8, 16, 32\}$, and $C_\mathbf{F}$ is the number of channels.

**Pixel Decoder.** A Transformer-based pixel decoder $\mathbf{Dec_P}$ is used to upsample $\mathbf{F}$ and gradually generate high-resolution per-pixel embeddings $\mathbf{P}$. $\mathbf{P}$ is then passed to the relationship decoder $\mathbf{Dec_R}$ to compute cross-attention with query features.

**Textual Encoder.** When a text prompt is provided for promptable VRS, we use the textual encoder $\mathbf{Enc_T}$ to encode it into a set of textual queries $\mathbf{Q^t} \in \mathbb{R}^{N_t \times C_q}$, where $N_t$ is the number of tokens in the textual queries, and $C_q$ denotes the channel number of query features. In practice, we use the textual encoder from CLIP [64] as $\mathbf{Enc_T}$. The format of the text prompt can be a single item (*e.g.* "*<p>predicate</p>*"), two of them ("*subject<p>predicate</p>*"), or all three of them, where "*predicate*" and "*subject*" denote category names of `predicate` and `subject`, respectively. "**", "*<o>*", "*<p>*" are used as separate tokens between `subject`, `predicate` and `object` in the text prompt. We could use natural language as the textual prompt instead of using a structured format. However, collecting the textual VRD data is not trivial, and we leave it as an extension of our model in future work.

**Relationship Decoder.** The relationship decoder $\mathbf{Dec_R}$, based on a Transformer decoder design, processes pixel decoder outputs $\mathbf{P}$ and latent queries $\mathbf{Q^v}$ to generate all possible triplets for standard VRS. Inside, masked attention [6] utilizes masks from earlier layers for foreground information. Each $\mathbf{Q^v}$ output feeds into five parallel heads: two mask heads for subject and object masks ($M_s, M_o$), two class heads for their categories ($C_s, C_o$), and another class head for relationship prediction ($C_p$) During training, Hungarian matching aligns predicted triplets with ground truth. For standard VRS inference, triplets above a confidence threshold are considered final predictions. For promptable VRS, $\mathbf{Enc_T}$ transforms text prompts into textual queries $\mathbf{Q^t}$ that are concatenated with $\mathbf{Q^v}$ and input into $\mathbf{Dec_R}$. This process, which uses self- and cross-attention mechanisms, generates `<subject, predicate, object>` triplets, similar to standard VRS. An additional matching loss during training ensures the model predicts triplets as specified by the text prompt. During inference, we calculate similarity scores between the textual query feature (last token's feature of $\mathbf{Q^t}$) and the latent query outputs. We then select entities and relationships specified in the textual prompt from the top $k$ triplets for the final outputs.

**Matching with textual features.** To enable open-vocabulary VRS, our `FleVRS` uses the CLIP textual encoder [64] to match visual features with candidate textual features for object and predicate

categories. We convert these categories into textual features using prompt templates, such as "*A photo of [predicate-ing]*" for HOI segmentation and "*A photo of something [predicate-ing] (something)*" for panoptic SGG.[1] The model computes matching scores between predicted class embeddings and these textual features, allowing classification beyond the fixed vocabulary of the training set and facilitating open-vocabulary VRS. Textual prompts are similarly encoded, and their features are used to calculate similarity scores for promptable VRS inference.

## 3.3 Loss functions

We use Hungarian matching during training to find the matched triplets with ground truth ones. For standard VRS, we compute focal losses $\mathcal{L}_b^s$, $\mathcal{L}_b^o$ and dice losses $\mathcal{L}_d^s$, $\mathcal{L}_d^o$ on `subject` and `object` mask predictions, cross-entropy losses $\mathcal{L}_c^s$, $\mathcal{L}_c^o$, $\mathcal{L}_c^p$ on `subject`, `object`, and `predicate` category classifications, which can be written as

$$\mathcal{L} = \lambda_b \sum_{i \in \{s,o\}} \mathcal{L}_b^i + \lambda_d \sum_{j \in \{s,o\}} \mathcal{L}_d^j + \sum_{k \in \{s,o,p\}} \lambda_c^k \mathcal{L}_c^k, \tag{1}$$

where $\lambda_b$, $\lambda_d$, and $\lambda_c$ are hyper-parameters for adjusting the weights of each loss. $\lambda_c^s$, $\lambda_c^o$, $\lambda_c^p$ are different classification loss weights for `subject`, `object`, and `predicate`. For promptable VRS, we adopt an additional matching loss $\mathcal{L}_g$ between the matched triplet class embedding and the textual query feature (the last token feature of $\mathbf{Q^t}$), which is in the form of cross-entropy loss. The final training loss is written as

$$\mathcal{L} = \lambda_b \sum_{i \in \{s,o\}} \mathcal{L}_b^i + \lambda_d \sum_{j \in \{s,o\}} \mathcal{L}_d^i + \sum_{k \in \{s,o,p\}} \lambda_c^k \mathcal{L}_c^k + \lambda_g \mathcal{L}_g, \tag{2}$$

where $\lambda_g$ controls the weight of $\mathcal{L}_g$. $\mathcal{L}_c$ depends on the text prompt. For example, given <`subject`, `predicate`>, there will not have $\mathcal{L}_c^s$ and $\mathcal{L}_c^p$ terms in Eq. (2), with `subject` and `predicate` categories being given. See the appendix for the concrete values of loss weights.

# 4 Experiments

## 4.1 Experimental Settings

**Datasets** For HOI segmentation, we utilize two public benchmarks: HICO-DET [4] and V-COCO [18]. To fit Our `FleVRS`, we use SAM [36] to transform box annotations into masks and apply Non-Maximum Suppression (NMS) to remove overlapping masks with an IoU threshold greater than 0.1. We omit `no_interaction` annotations from HICO-DET due to incomplete annotation, leaving 44,329 images (35,801 training, 8,528 testing) with 520 HOI classes from 80 objects and 116 actions.[2] V-COCO is built from COCO [49], comprising 10,396 images (5,400 training, 4,964 testing), featuring 80 objects and 29 actions, and includes 263 HOI classes. Both datasets align with COCO's object categories. For panoptic SGG, we use the PSG dataset [85], sourced from COCO and VG [37] intersections, containing 48,749 images (46,572 training, 2,177 testing) with 133 objects and 56 predicates.

**Data Structure for open-vocabulary HOI segmentation** Following prior studies [3, 23], we evaluate HICO-DET under three scenarios: (1) Unseen Composition (UC), where some HOI classes are absent despite all object and verb categories being present; (2) Unseen Object (UO), where certain object classes and their corresponding HOI triplets are excluded from training; and (3) Unseen Verb (UV), where specific verb classes and their associated triplets are similarly omitted. In UC, the Rare First (RF-UC) approach targets tail HOI classes, while Non-rare First (NF-UC) focuses on head categories. Originally, UC included 120/480/600 categories for unseen/seen/full sets, which reduces to 115/405/520 after removing `no_interaction` annotations. For UO, we select 12 unseen objects from 80, resulting in 88/432 unseen/seen HOI categories.

**Evaluation Metric** For standard HOI segmentation, we convert the predicted masks to bounding boxes to compare with current methods, and follow the setting in [4] to use the mean Average Precision (mAP) for evaluation. We also turn the outputs of other methods into masks and report

---

[1]Omit "*something*" for spatial relationships.
[2]interchangeable with "verb", "predicate".

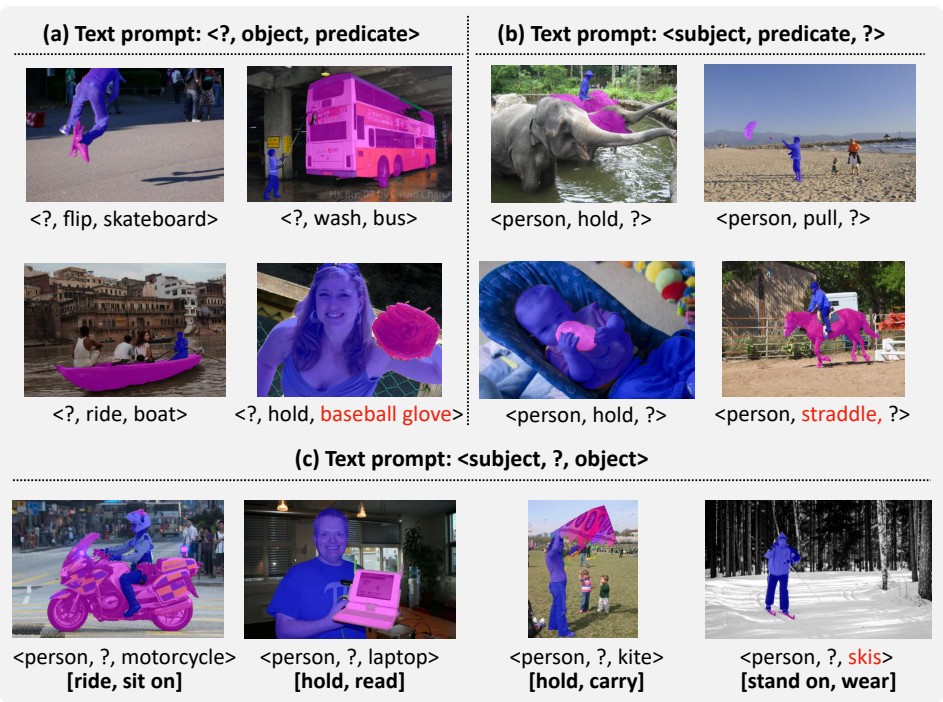

**(a) Text prompt: <?, object, predicate>**

<?, flip, skateboard>

<?, wash, bus>

<?, ride, boat>

<?, hold, baseball glove>

**(b) Text prompt: <subject, predicate, ?>**

<person, hold, ?>

<person, pull, ?>

<person, hold, ?>

<person, straddle, ?>

**(c) Text prompt: <subject, ?, object>**

<person, ?, motorcycle>
**[ride, sit on]**

<person, ?, laptop>
**[hold, read]**

<person, ?, kite>
**[hold, carry]**

<person, ?, skis>
**[stand on, wear]**

Figure 4: **Qualitative results of promptable VRS on HICO-DET [4] test set.** We show visualizations of `subject` and `object` masks and relationship category outputs, given three types of text prompts. In (c), we show the predicted predicates in bold characters. Unseen objects and predicates are denoted in red characters.

mask $mAP$ for thorough comparison. An HOI triplet prediction is a true positive if (1) both predicted human and object bounding boxes/masks have IoU larger than 0.5 *w.r.t.* GT boxes/masks; (2) Both the predicted object and verb categories are correct. For HICO-DET, we evaluate the three different category sets: all 520 HOI categories (Full), 112 HOI categories (less than 10 training instances) (Rare), and the other 408 HOI categories (Non-Rare). For VCOCO, we report the role mAPs in two scenarios: (1) S1: 29 actions including 4 body motions; (2) S2: 25 actions without the no-object HOI categories. For standard panoptic SGG, following [85], we use $R@K$ and $mR@K$ metrics, which calculate the triplet recall and mean recall for every predicate category, given the top K triplets from the model. A successful recall requires both subject and object to have mask-based IoU larger than 0.5 compared to their GT masks, with the correct predicate classification in the triplet.

**Implementation Details** Following [6, 109], we use 100 latent queries and 9 decoder layers in the relationship decoder. We adopt Focal-T/L [84] for the Image Encoder and DaViT-B/L for the pixel decoder. We use the textual encoder from CLIP to encode input text prompt and subject, object, and predicate categories. During training, we set the input image to be $640 \times 640$, with batch size of 64. We optimize our network with AdamW [54] with a weight decay of $10^{-4}$. We train all models for 30 epochs with an initial learning rate of $10^{-4}$ decreased by 10 times at the 20th epoch. To improve training efficiency, we initialize Our `FleVRS` using the pre-trained weights from [109]. For all experiments, the parameters of the textual encoder are frozen except its logit scales. The loss weights $\lambda_b$, $\lambda_d$, $\lambda_c$ and $\lambda_{grd}$ (superscript omitted) are set to 1,1,2, and 2. More details are in the appendix.

### 4.2 Standard VRS

We evaluate our method on three benchmarks, *i.e.* HICO-DET [4], VCOCO [18] for HOI segmentation, and PSG [85] for the panoptic SGG.

**HOI segmentation** Since Our `FleVRS` leverages mask supervision, either converting mask results into bounding boxes or transforming bounding boxes from previous methods' output into masks does not facilitate a completely equitable comparison. For the utmost fairness in comparison, we report both box $mAP$ and mask $mAP$ from the above ways. As shown in Table 2, Our `FleVRS` shows superior

| Model | Backbone | Default (%) | | |
|---|---|---|---|---|
| | | box/mask mAP$_F$ | box/mask mAP$_R$ | box/mask mAP$_N$ |
| *Bottom-up methods* | | | | |
| SCG [96] | ResNet-50 | 31.3 / 31.3 | 24.7 / 25.0 | 33.3 / 35.5 |
| UPT [97] | ResNet-101 | 32.6 / 34.9 | 28.6 / 29.4 | 33.8 / 36.1 |
| STIP [100] | ResNet-50 | 32.2 / 30.8 | 28.2 /28.6 | 33.4 / 32.5 |
| ViPLO [60] | ViT-B | 37.2 / 39.1 | 35.5 / 37.8 | 37.8 / 39.7 |
| *Additional training with object detection data* | | | | |
| UniVRD [101] | ViT-L | 37.4 / - | 28.9 / - | 39.9 / - |
| PViC [98] | Swin-L | 44.3 / - | 44.6 / - | 44.2 / - |
| RLIPv2 [92] | Swin-L | 45.1 / 48.6 | 45.6 / 44.3 | 43.2 / 49.8 |
| *Single-stage methods* | | | | |
| HOTR [33] | ResNet-50 | 25.1 / 26.5 | 17.3 / 18.5 | 27.4 / 29.0 |
| QPIC [70] | ResNet-101 | 29.9 / 30.5 | 23.0 / 23.1 | 31.7 / 33.1 |
| CDN [95] | ResNet-101 | 32.1 / 33.9 | 27.2 / 28.9 | 33.5 / 36.0 |
| RLIP [91](VG+COCO) | ResNet-50 | 32.8 / 34.4 | 26.9 / 27.7 | 34.6 / 36.5 |
| GEN-VLKT [47] | ResNet-101 | 35.0 / 35.6 | 31.2 / 32.6 | 36.1 / 37.8 |
| ERNet [48] | EfficientNetV2-XL | 35.9 / - | 30.1 / - | 38.3 / - |
| MUREN [35] | ResNet-50 | 32.9 / 35.4 | 28.7 / 30.1 | 34.1 / 37.6 |
| **Ours** | Focal-L | **38.1 / 40.5** | **33.0 / 34.9** | **39.5 / 42.4** |

Table 2: **Quantitative results on the HICO-DET test set.** We report both box and mask $mAP$ under the *Default* setting [4] containing the *Full* (F), *Rare* (R), and *Non-Rare* (N) sets. `no_interaction` class is removed in mask mAP. The best score is highlighted in bold, and the second-best score is underscored. '-' means the model did not release weights and we cannot get the mask $mAP$. Due to space limit, we show the complete table with more models in the appendix.

| Model | Backbone | AP$_{role}^{S\#1}$ | AP$_{role}^{S\#2}$ |
|---|---|---|---|
| *Bottom-up methods* | | | |
| VSGNet [72] | ResNet-152 | 51.8 / - | 57.0 / - |
| ACP [34] | ResNet-152 | 53.2 / - | - / - |
| IDN [43] | ResNet-50 | 53.3 / - | 60.3 / - |
| STIP [100] | ResNet-50 | **66.0** / 66.2 | **70.7 / 70.5** |
| *Additional training with object detection data* | | | |
| UniVRD [101] | ViT-L | 65.1 / - | 66.3 / - |
| PViC [98] | Swin-L | 64.1 / - | 70.2 / - |
| RLIPv2 [92] | Swin-L | 72.1 / 71.7 | 74.1 / 73.5 |
| *Single-stage methods* | | | |
| HOTR [33] | ResNet-50 | 55.2 / 55.0 | 64.4 / 64.1 |
| DIRV [11] | EfficientDet-d3 | 56.1 / - | - / - |
| CDN [95] | ResNet-101 | 63.9 / 61.3 | 65.8 / 63.2 |
| RLIP [91] | ResNet-50 | 61.9 / 61.3 | 64.2 / 64.0 |
| GEN-VLKT [47] | ResNet-101 | 63.6 / 61.8 | 65.9 / 64.0 |
| ERNet [48] | EfficientNetV2-XL | 64.2 / - | - / - |
| **Ours** | Focal-L | 65.2 / **66.5** | 66.5 / 67.9 |

Table 3: **Quantitative results on V-COCO.** We report both box and mask $mAP$. The best score is highlighted in bold, and the second-best score is underscored. '-' means the model did not release weights and we cannot get the mask $mAP$. Due to space limit, we show the complete table with more models in the appendix.

performance over current single-stage methods in terms of box and mask $mAP$ on HICO-DET. We also achieve competitive performance on VCOCO [18], as shown in Table 3. The advantages of Our `FleVRS` come from: (1) one-stage Transformer-based design with fine-grained training supervision for VRS. With subject and object masks, the model has more accurate supervision, compared with box annotations that contain redundancy [85]. (2) good language-visual alignment with the large-scale pretrained model [64]. Our `FleVRS` achieves competitive results without additional training on large-scale detection datasets [101]. Among one-stage HOI methods, our approach is simpler and able to tackle different datasets without modifications to the structure.

**Panoptic SGG** From Table 4, Our `FleVRS` can achieve competitive results in terms of $R@50$ and $R@100$ without elaborated designs for PSG, compared with most of previous work. Our `FleVRS` is not superior to HiLo [104], which is mainly due to the long-tail distribution of the dataset and the

| Method | Backbone | R/mR@20 | R/mR@50 | R/mR@100 |
|---|---|---|---|---|
| *Adapted from SGG methods* | | | | |
| IMP [81] | VGG-16 | 17.9 / 7.35 | 19.5 / 7.88 | 20.1 / 8.02 |
| MOTIFS [93] | VGG-16 | 20.9 / 9.60 | 22.5 / 10.1 | 23.1 / 10.3 |
| VCTree [71] | VGG-16 | 21.7 / 9.68 | 23.3 / 10.2 | 23.7 / 10.3 |
| GPSNet [50] | VGG-16 | 18.4 / 6.52 | 20.0 / 6.97 | 20.6 / 7.17 |
| *One-stage PSG methods* | | | | |
| PSGTR [85] | ResNet-101 | **28.2 / 15.4** | **32.1 / 20.3** | **35.3 / 21.5** |
| PSGFormer [85] | ResNet-101 | 18.0 / 14.2 | 20.1 / 18.3 | 21.0 / 19.8 |
| *Training with additional data* | | | | |
| HiLo [104] | Swin-L | 40.6 / 29.7 | 48.7 / 37.6 | 51.4 / 40.9 |
| **Ours** | Focal-L | 27.0 / 15.4 | 31.0 / 18.3 | 31.7 / 18.8 |

Table 4: **Quantitative results on PSG.** The best score is highlighted in bold, and the second-best score is underscored.

limitation of using CLIP to encode abstract relationships (*e.g.*, entering, exiting). The model tends to predict high-frequency relationships and is hard to understand and predict low-frequency ones.

**Ablation study.** We ablate Our `FleVRS` by testing different encoding strategies for relationships via the textual encoder in 7. Specifically, we compare encoding `object` and `predicate` categories as `<person, predicate, object>` triplets or separately, associating the results with either triplet cross-entropy (CE) loss or disentangled CE loss. Results reveal that while HICO-DET benefits from the disentangled CE loss, allowing better generalization to novel concepts, VCOCO performs better with triplet CE loss due to the challenge of distinguishing verbs without corresponding objects in various contexts (*e.g.*, differentiating "eat" in "a person eating an apple" *vs* "a person eating"). Further experiments with various backbones demonstrate performance enhancements with larger models. Additionally, incorporating a box head for supervision alongside mask supervision enhances performance, which is attributed to the masked attention mechanism inspired by [6]. Exploring the potential synergies of training across multiple datasets, we find that while unified training improves VCOCO's performance due to its smaller size, HICO-DET and PSG show limited gains. This disparity is likely due to the different predicate categories used in PSG compared to HICO-DET and VCOCO.

**Comparison with previous works.** UniVRD uses a two-stage approach, where the model first detects independent objects and then decodes relationships between them, retrieving boxes from the initial detection stage. In contrast, our method employs a one-stage approach, where each query directly corresponds to a <subject, object, predicate> triplet. This transition improves time efficiency from O(MxN) to O(K), where M is the number of subject boxes, N is the number of object boxes, and K is the number of interactive pairs. Our approach also provides greater flexibility by learning a unified representation that encompasses object detection, subject-object association, and relationship classification in a single model.

In terms of training data, we use much fewer training data (x50 less, without using VG [37] and Objects365 [66]) and Our `FleVRS` with the Focal-L [84] backbone is much smaller than UniVRD [101] (164M vs 640M) with LiT(ViT-H/14), we achieve comparable results(37.4 vs 38.1 on HICO-DET). While our method does not match RLIPv2 [92] in performance, this is due to different design philosophies and goals. RLIPv2 is a two-stage approach optimized for large-scale pre-training and relies on separately trained detectors. Our `FleVRS`, however, is not designed for pretraining and does not include a separately trained detector. Our focus is on enhancing the flexibility of the VRS

| | No subject | No object | Only predicate | |
| | S-IoU | O-IoU | S-IoU | O-IoU |
|---|---|---|---|---|
| *Conv-based methods* | | | | |
| VRD [55] | 0.208 | 0.008 | 0.024 | 0.026 |
| SSAS [38] | 0.335 | 0.363 | 0.334 | 0.365 |
| **Ours** | **0.568** | **0.364** | **0.556** | **0.366** |

Table 5: Comparison of promptable VRD results with the baseline on VRD dataset [55].

model without *directly* training on extensive curated data(x50 more, VG and Objects365). Thus, the differences in performance are attributed to the scale and design objectives. We further discuss the FLOPs and the number parameters of the backbone compared to previous works in the Appendix.

### 4.3 Promptable VRS

We evaluate the ability of promptable VRS on the VRD dataset [55], to compare with [38]. As in Table 5, Our `FleVRS` can locate entities given flexible text query inputs and performs better localizing

subjects and objects. Our `FleVRS` gets particular better results on localizing subjects (0.568 *vs* 0.335, 0.556 *vs* 0.334), which is mainly because there are fewer categories in subjects compared with objects and lots of subjects are humans, making it easier to segment subjects. We further evaluate our promptable VRS approach on HICO-DET and PSG, as they contain rich relationship labels. Since there are no previous baselines, we show qualitative results in Fig. 4. We visualize the subject and object masks with the highest matching score for each example. We can see that the model is able to localize `subject` and `object` masks and predict their relationships given the structured textual prompt. We further perform postprocessing way to search triplets from standard VRS output, which serves as another baseline to show the effectiveness of our method. Please refer to section E in the appendix for results and discussions of fair comparison.

**Difference with standard REC tasks** The referring expression comprehension (REC) tasks on benchmarks like RefCOCO [30], RefCOCO+ [59], and RefCOCOg [90] are designed to detect objects based on free-form textual phrases, such as "a ball and a cat" or "Two pandas lie on a climber." In contrast, the promptable VRS task in our work focuses on detecting subject-object pairs within a structured prompt format, such as `<?, sit_on, bench>` or `<person, ?, horse>`, as illustrated in Fig. 1 of the main paper. Our `FleVRS` is designed to encode and compute similarity scores for each of these elements separately. Our primary focus is on relational object segmentation based on a single structured query, which differs significantly from the objectives of REC benchmarks.

## 4.4 Open-vocabulary VRS

We conduct open-vocabulary experiments following the defined zero-shot HOI detection setting [23, 25, 26, 47] on HICO-DET. As shown in Table 6, Our `FleVRS` surpasses previous single-dataset methods across all settings, with its open-vocabulary capabilities stemming from the knowledge transferred from CLIP [64]. GEN-VLKT [47] also leverages CLIP to facilitate open-vocabulary capabilities by encoding `<person, predicate, object>` as a triplet and using it for HOI category classification. In contrast, our approach separates the encoding of `predicate` and `object`, enhancing the model's generalization ability over novel concepts.

| Method | Unseen | Seen | Full |
|---|---|---|---|
| *Rare First Unseen Composition* | | | |
| VCL [24] | 10.06 | 24.28 | 21.43 |
| ATL [25] | 9.18 | 24.67 | 21.57 |
| FCL [26] | 13.16 | 24.23 | 22.01 |
| GEN-VLKT [47] | 21.36 | 32.91 | 30.56 |
| **Ours** | **26.06** | **39.61** | **36.60** |
| *Non-rare First Unseen Composition* | | | |
| VCL [24] | 16.22 | 18.52 | 18.06 |
| ATL [25] | 18.25 | 18.78 | 18.67 |
| FCL [26] | 18.66 | 19.55 | 19.37 |
| GEN-VLKT [47] | 25.05 | 23.38 | 23.71 |
| **Ours** | **26.62** | **31.17** | **30.17** |
| *Unseen Object* | | | |
| FCL [26] | 0.00 | 13.71 | 11.43 |
| ATL [25] | 5.05 | 14.69 | 13.08 |
| GEN-VLKT [47] | 10.51 | 28.92 | 25.63 |
| **Ours** | **14.48** | **35.28** | **31.71** |
| *Unseen Verb* | | | |
| GEN-VLKT [47] | 20.96 | 30.23 | 28.74 |
| **Ours** | **21.50** | **35.63** | **33.09** |

Table 6: Results of open-vocabulary HOI detection on HICO-DET.

| Variant | HICO-DET mask mAP$_F$ | VCOCO mask AP$_{role}^{S\#1}$ | PSG R/mR@20 |
|---|---|---|---|
| *Different losses* | | | |
| Disentangled CE loss | **40.5** | 62.1 | **27.0 / 15.4** |
| Triplet CE loss | 36.8 | **66.5** | 25.5 / 14.6 |
| Disentangled CE loss + Triplet CE loss | 39.0 | 64.5 | 26.5 / 14.8 |
| *Different visual backbones* | | | |
| Focal Tiny | 34.2 | 59.8 | 25.8 / 15.0 |
| Focal Large | **40.5** | **66.5** | **27.0 / 15.4** |
| *Different design choices* | | | |
| Box head only | 33.0 | 62.0 | - |
| Mask head only | 40.5 | 66.5 | 27.0 / 15.4 |
| Mask and box head | **41.2** | **67.0** | - |
| *Different training datasets* | | | |
| Single source | **40.5** | 66.5 | 27.0 |
| HICO-DET+VCOCO | 40.3 | **66.9** | - |
| HICO-DET+VCOCO+PSG | 40.0 | 66.4 | **27.6** |

Table 7: Ablations of different loss types, backbones, design choices and training sets. We adopt the Focal-L backbone by default.

## 5 Conclusion

In this work, we present a novel approach for visual relationship segmentation that integrates the three critical aspects of a flexible VRS model: standard VRS, promptable querying, and open-vocabulary capabilities. Our `FleVRS` demonstrates the ability to not only support HOI segmentation and panoptic SGG but also to do so in response to various textual prompts and across a spectrum of previously unseen objects and interactions. By harnessing the synergistic potential of textual and visual features, our model delivers promising experimental results on existing benchmark datasets. We hope our work can serve as a solid stepping stone for pursuing more flexible visual relationship segmentation models.

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

# A  Limitations and Future Work

Deploying our model in real-world scenarios requires specialized pretraining data for relationship understanding, which is notably scarce. The lack of automated annotation pipelines and dependence on the CLIP model pose scalability challenges due to specific resource requirements. Ideally, we aim for a single general-purpose framework that can be trained on multiple datasets and enhance performance across various tasks and benchmarks. However, achieving this remains a challenge with our current model. We leave the exploration of how to synergize different datasets and develop effective training strategies to future work. While integrating free-form text inputs is more natural as large language models evolve, it necessitates additional preprocessing to align with our framework. Furthermore, the absence of comparable methods for promptable VRS makes complete fair benchmarking difficult.

# B  Model Structure Details

We use Focal T/L [84] network as the image encoder $\mathbf{Enc_I}$. Given the image $\mathbf{I} \in \mathbb{R}^{H \times W \times 3}$, we pass it to $\mathbf{Enc_I}$ and obtain multi-scale features of different strides and channals $\mathbf{F} = \{\mathbf{F}_s | s = 4, 8, 16, 32\}$, where $s$ is the stride.

Then, the pixel decoder $\mathbf{Dec_p}$ gradually upsample $\mathbf{F}$ to generate high-resolution per-pixel embeddings $\mathbf{P} = \{\mathbf{P}_i | i = 1, 2, 3, 4\}$, where $i$ is the layer number and different $\mathbf{P}_i$s have the same channel number but different resolutions. $\mathbf{P}$ will then input to $\mathbf{Dec_R}$.

Under standard VRS, $\mathbf{Dec_R}$ takes latent queries $\mathbf{Q^v}$ and $\mathbf{P}$ as inputs. Under promptable VRS, we use the textual encoder $\mathbf{Enc_T}$ to encode the textual prompt into a set of textual queries $\mathbf{Q^t}$ and concatenate $\mathbf{Q^t}$ with the latent queries $\mathbf{Q^v}$, and input them to $\mathbf{Dec_R}$. Inside $\mathbf{Dec_R}$, cross- and self-attention are computed among queries and per-pixel embeddings, where masked attention [6] is adopted to enhance the foreground regions of predicted masks.

On top of latent queries output $\mathbf{Q}_o^v \in \mathbb{R}^{N_v \times C_q}$ ($N_v$ is the number of latent queries, $C_q$ is the channel number), there are five heads, producing predictions in parallel. They are two mask heads $f_{M_s}(\cdot)$, $f_{M_o}(\cdot)$ for predicting subject and object masks ($M_s$, $M_o$), and two class heads $g_{C_s}(\cdot)$, $g_{C_o}(\cdot)$ for predicting their object categories ($C_s$, $C_o$). Another class head $g_{C_p}$ is used to predict relationships $C_p$ for this `<subject, object>` pair. Detailed operations can be written as

$$M_s = \text{Up}\left[\mathbf{P}_4 \cdot f_{M_s}(\mathbf{Q}_o^v)\right], \tag{3}$$
$$M_o = \text{Up}\left[\mathbf{P}_4 \cdot f_{M_o}(\mathbf{Q}_o^v)\right], \tag{4}$$
$$C_s = \mathbf{T}_s \cdot g_{C_s}(\mathbf{Q}_o^v), \tag{5}$$
$$C_o = \mathbf{T}_o \cdot g_{C_o}(\mathbf{Q}_o^v), \tag{6}$$
$$C_p = \mathbf{T}_p \cdot g_{C_p}(\mathbf{Q}_o^v), \tag{7}$$

where $\text{Up}[\cdot]$ denotes the upsampling operation, $\mathbf{T}_s$, $\mathbf{T}_o$ and $\mathbf{T}_p$ denote candidate textual features of subject, object and predicate categories that are encoded by CLIP [64]. The mask embeddings $f_{M_s}(\mathbf{Q}_o^v)$ and $f_{M_o}(\mathbf{Q}_o^v)$ compute the dot products with the last layer's per-pixel embedding $\mathbf{P}_4$, respectively, and upsample to the original resolution as final mask predictions.

**Training.** We employ Hungarian matching to align predicted triplets with ground truth, calculating mask and category classification losses on these matches. For promptable VRS, we introduce a matching loss to assess the similarity between the matched triplet embedding and the textual prompt's feature, formulated as a cross-entropy loss. The triplet embedding combines class embeddings from class heads. For the textual prompt's feature, we use the last token's feature from textual queries $\mathbf{Q^t}$. For instance, with a textual prompt like `<subject, predicate, ?>`, where the subject and predicate are specified, the similarity measurement utilizes the summation of their class embeddings $g_{C_s}(\mathbf{Q}_o^v) + g_{C_p}(\mathbf{Q}_o^v)$.

**Inference.** Under standard VRS, we compute the confidence score of each triplet, which comes from the product of subject, object, and predicate classification scores. We take top $k_s$ ($k_s = 100$) triplets to compute mean average precision for HOI segmentation and mean recall for panoptic SGG. Under promptable VRS, we compute similarities between the textual prompt's feature and triplet embeddings and choose $k_f$ ($k_f = 10$) as the final predicted triplets.

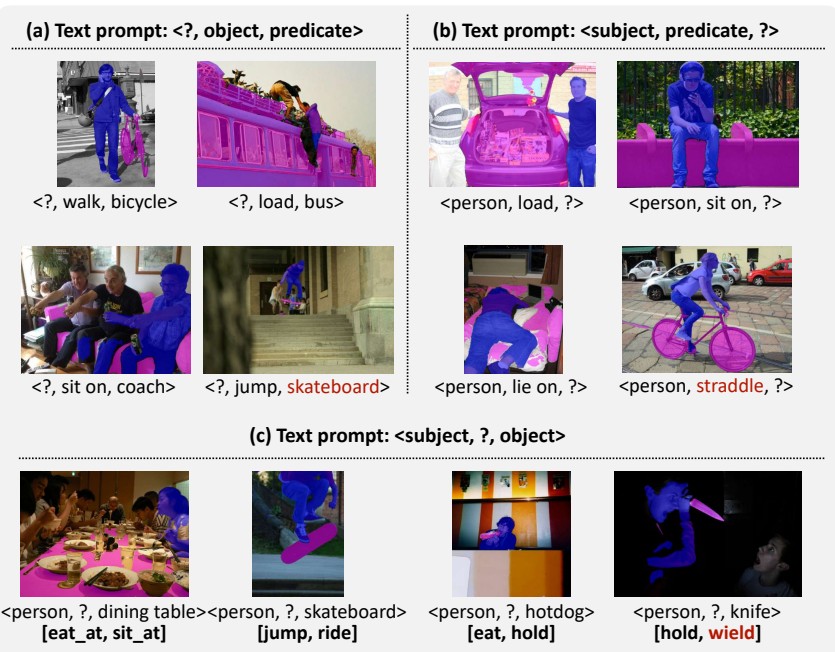

**(a) Text prompt: <?, object, predicate>**

<?, walk, bicycle>  <?, load, bus>

<?, sit on, coach>  <?, jump, skateboard>

**(b) Text prompt: <subject, predicate, ?>**

<person, load, ?>  <person, sit on, ?>

<person, lie on, ?>  <person, straddle, ?>

**(c) Text prompt: <subject, ?, object>**

<person, ?, dining table>  <person, ?, skateboard>  <person, ?, hotdog>  <person, ?, knife>
**[eat_at, sit_at]**  **[jump, ride]**  **[eat, hold]**  **[hold, wield]**

Figure 5: **Qualitative results of promptable and open-vocabulary VRS on HICO-DET [4] test set.** We show visualizations of the predicted triplet with the highest matching score, including subject, object masks, and predicted predicate categories. There are three types of textual prompts shown in (a), (b), and (c), with unseen concepts in the rightmost columns. In (c), we show the predicted predicates in bold characters. Unseen objects and predicates are denoted in red characters. Note that the subject is always "person" in HICO-DET.

## C  Implementation details

We set the input image size to be $640 \times 640$, with batch size as 64. The model is optimized with AdamW [54] with a weight decay of $10^{-4}$. We set $N_v = 100$ and $N_v = 200$ for Focal-T and Focal-L backbones, respectively, and $C_q = 512$. The structure of pixel decoder $\mathbf{Dec_p}$ is a Transformer encoder with 6 encoder layers and 8 heads. The structure of relationship decoder $\mathbf{Dec_R}$ is a Transformer decoder with 9 decoder layers.

**Standard HOI segmentation** The model only takes the image as input without textual prompts. Since the subject class is always "person" in HOI segmentation, we omit the subject class head. The model is trained with 30 epochs, with an initial learning rate of $10^{-4}$ ($10^{-5}$ for the image encoder) decreased by 10 times at the 20th epoch. The loss weights $\lambda_b$, $\lambda_d$, $\lambda_c^o$ and $\lambda_c^p$ are set to be 2, 1, 1, 2.

**Promptable HOI segmentation** To enable promptable HOI segmentation, we build three types of textual prompts: 1) "$person<p>predicate</p>$"; 2) "$<p>predicate</p><o>object</o>$"; 3) "$person<o>object</o>$". We evaluate the model on HICO-DET [4] since it contains richer human-object interactions than VCOCO [18]. During training, we randomly sample various types of text prompts and simultaneously train different objectives using distinct loss terms. To prevent the model from learning shortcuts, we select one ground truth triplet per training image and pair it with a randomly chosen textual prompt type. This approach ensures a balanced distribution of labeled training data for promptable VRS across different prompt types.

**Standard panoptic SGG** We use the subject class head to predict the subject category and the model does not have textual prompts as inputs. The model is trained with 60 epochs, with an initial learning rate of $10^{-4}$ ($10^{-5}$ for the image encoder) decreased by 10 times at the 40th epoch. The loss weights $\lambda_b$, $\lambda_d$, $\lambda_c^s$, $\lambda_c^o$ and $\lambda_c^p$ are set to be 2, 1, 1, 1, 2.

**Promptable panoptic SGG** Similar to promptable HOI segmentation, there are three types of textual prompts: 1) "$subject<p>predicate</p>$"; 2) "$<p>predicate</p><o>object</o>$"; 3) "$subject<o>object</o>$". Similarly, during training, we randomly

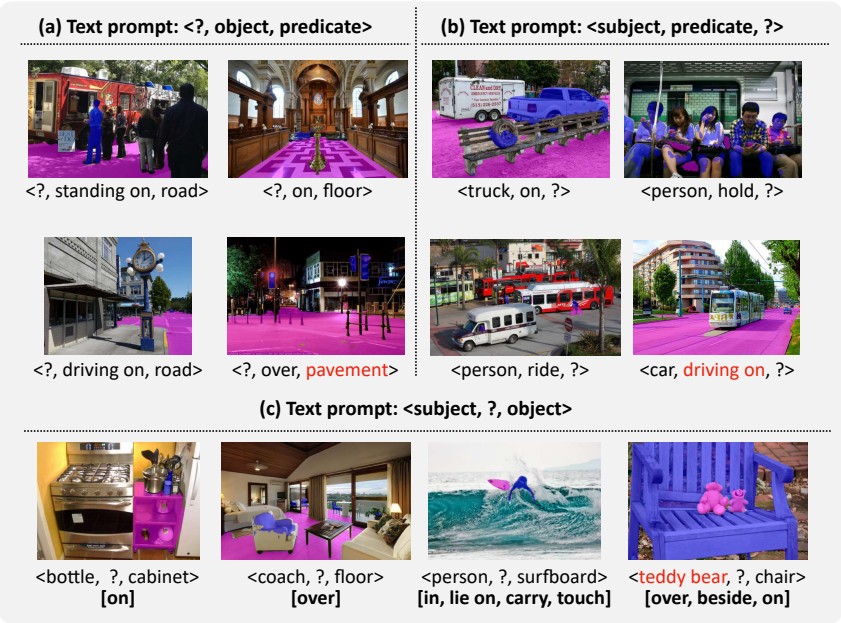

Figure 6: **Qualitative results of promptable and open-vocabulary VRS on PSG [85] test set.** We show visualizations of the predicted triplet with the highest matching score, including subject, object masks, and predicted predicate categories. There are three types of textual prompts shown in (a), (b), and (c), with unseen concepts in the rightmost columns. In (c), we show the predicted predicates in bold characters. Unseen objects and predicates are denoted in red characters.

sample different types of text prompts, and different objectives are trained simultaneously with different loss terms. We also keep a balanced distribution of labeled training data for panoptic SGG. We set the weight of grounding loss $\lambda_g$ to be 2, while other weights are the same as standard panoptic SGG.

**Open-vocabulary promptable VRS** We adopt the zero-shot setting from [47] for open-vocabulary HOI segmentation. To streamline our approach, we integrate the open-vocabulary promptable setting within the broader context of open-vocabulary VRS. In this setting, 'open-vocabulary' refers to handling both seen and unseen categories in the input textual prompts. We randomly exclude object and predicate categories during training and assess our model on these as well as on seen categories. Given the absence of existing benchmarks for this specific challenge, we present qualitative results demonstrating our model's proficiency in open-vocabulary promptable VRS.

## D  Qualitative results of promptable and open-vocabulary VRS

We show qualitative results of promptable and open-vocabulary VRS on HOI segmentation and panoptic SGG by giving the model different types of structured textual prompts. For simplicity, we show examples of omitting only one component of the triplet. We can see that our model is able to localize the correct subject and object and complement the missing element corresponding to the given textual prompt, *e.g.* <person, ?, dining_table> in Fig. 5(c) and <truck, on, ?> in Fig. 6(b). The model can also predict multiple interactions for the same subject-object pair, as shown in Fig. 5(c) and Fig. 6(c). We further trained two versions by removing unseen objects and unseen predicates, respectively. We show that our model can detect novel objects and predicates by feeding unseen concepts in textual prompts, as in the rightmost columns of Fig. 5 and Fig. 6. Fig. 6(b) shows the model outputs multiple instances in one subject mask due to similar patterns occurring in the training set. Note that the flexible VRD task is more difficult on the PSG [85] dataset due to its complexity of scenes, while we make the first attempt and our model is still able to show promising grounding results.

| Model | Backbone | Default (%) | | |
|---|---|---|---|---|
| | | box/mask mAP$_F$ | box/mask mAP$_R$ | box/mask mAP$_N$ |
| *Bottom-up methods* | | | | |
| InteractNet [14] | ResNet-50 | 9.9 / - | 7.2 / - | 10.8 / - |
| iCAN [12] | ResNet-50 | 14.8 / - | 10.5 / - | 16.2 / - |
| No-Frills [19] | ResNet-152 | 17.2 / - | 12.2 / - | 18.7 / - |
| DRG [13] | ResNet-50 | 24.5 / - | 19.5 / - | 26.0 / - |
| VSGNet [72] | ResNet-152 | 19.8 / - | 16.1 / - | 20.9 / - |
| FCMNet [53] | ResNet-50 | 20.4 / - | 17.3 / - | 21.6 / - |
| IDN [43] | ResNet-50 | 23.4 / - | 22.5 / - | 23.6 / - |
| ATL [25] | ResNet-101 | 23.8 / - | 17.4 / - | 25.7 / - |
| SCG [96] | ResNet-50 | 31.3 / 31.3 | 24.7 / 25.0 | 33.3 / 35.5 |
| UPT [97] | ResNet-101 | 32.6 / 34.9 | 28.6 / 29.4 | 33.8 / 36.1 |
| STIP [100] | ResNet-50 | 32.2 / 30.8 | 28.2 /28.6 | 33.4 / 32.5 |
| ViPLO [60] | ViT-B | 37.2 / 39.1 | 35.5 / 37.8 | 37.8 / 39.7 |
| *Additional training with object detection data* | | | | |
| UniVRD [101] | ViT-L | 37.4 / - | 28.9 / - | 39.9 / - |
| PViC [98] | Swin-L | 44.3 / - | 44.6 / - | 44.2 / - |
| RLIPv2 [92] | Swin-L | 45.1 / 48.6 | 45.6 / 44.3 | 43.2 / 49.8 |
| *Single-stage methods* | | | | |
| DIRV [11] | EfficientDet-d3 | 21.8 / - | 16.4 / - | 23.4 / - |
| PPDM-Hourglass [46] | DLA-34 | 21.9 / - | 13.9 / - | 24.3 / - |
| HOI-Transformer [108] | ResNet-101 | 26.6 / - | 19.2 / - | 28.8 / - |
| GGNet [103] | Hourglass-104 | 29.2 | 22.1 / - | 30.8 / - |
| HOTR [33] | ResNet-50 | 25.1 / 26.5 | 17.3 / 18.5 | 27.4 / 29.0 |
| QPIC [70] | ResNet-101 | 29.9 / 30.5 | 23.0 / 23.1 | 31.7 / 33.1 |
| CDN [95] | ResNet-101 | 32.1 / 33.9 | 27.2 / 28.9 | 33.5 / 36.0 |
| RLIP [91](VG+COCO) | ResNet-50 | 32.8 / 34.4 | 26.9 / 27.7 | 34.6 / 36.5 |
| GEN-VLKT [47] | ResNet-101 | 35.0 / 35.6 | 31.2 / 32.6 | 36.1 / 37.8 |
| ERNet [48] | EfficientNetV2-XL | 35.9 / - | 30.1 / - | 38.3 / - |
| MUREN [35] | ResNet-50 | 32.9 / 35.4 | 28.7 / 30.1 | 34.1 / 37.6 |
| **Ours** | Focal-L | **38.1 / 40.5** | **33.0 / 34.9** | **39.5 / 42.4** |

Table 8: **Quantitative results on the HICO-DET test set.** We report both box and mask $mAP$ under the *Default* setting [4] containing the *Full* (F), *Rare* (R), and *Non-Rare* (N) sets. `no_interaction` class is removed in mask mAP. The best score is highlighted in bold, and the second-best score is underscored. '-' means the model did not release weights and we cannot get the mask $mAP$.

## E    Quantitative results of standard VRS

Due to the large number of works on HOI detection, we show the complete comparison with previous methods in Fig. 8 and Fig. 9. Our model achieves competitive results on both datasets, especially compared with other single-stage methods. From Table 9, MUREN [35] gets the best result on VCOCO (68.8 *vs.* 65.2, 68.2 *vs.* 66.5), but cannot achieve a similarly strong result on HICO-DET (32.9 *vs.* 38.1, 35.4 *vs.* 40.5), where the verb categories are more complicated.

**Fair Comparison.** Since existing models use bounding box annotations to train and evaluate $mAP$, we ensure fair comparisons by converting our model's output masks into bounding boxes to compute box $mAP$. Additionally, we apply released weights from previous methods, transform their output boxes into segmentation masks using SAM [36], and report mask $mAP$. In both metrics, our model demonstrates superior performance.

We further train the existing HOI detectors CDN [95], STIP, GEN-VLKT the same SAM generated data used in our paper, which leads to worse accuracy on HICO-DET, as in Tab. 10. Thus, the major performance improvements of our work are due to both the SAM-labeled data and our architectural design.

At the same time, we also train our model with bounding boxes only, where we get decreased accuracy ($mAP$ of 30.7 *vs* 36.3). We attribute it to the network architecture derived from Mask2Former [6], which is mainly designed for pixel-wise segmentation tasks.

**FLOPs and the number parameters of the backbone compared to previous works.** As in Table 2 and 3 of the main paper, we have done extensive comparisons with previous methods, including backbones on ResNet-50/101/152, EfficientNet, Hourglass, Swin Transformers, and LiT architectures. For previous methods that utilize ResNet backbone for HOI detection and PSG, our comparison

| Model | Backbone | $AP_{role}^{S\#1}$ | $AP_{role}^{S\#2}$ |
|---|---|---|---|
| *Bottom-up methods* | | | |
| InteractNet [14] | ResNet-50 | 40.0 / - | - / - |
| GPNN [62] | ResNet-50 | 44.0 / - | - / - |
| iCAN [12] | ResNet-50 | 45.3 / - | 52.4 / - |
| TIN [42] | ResNet-50 | 47.8 / - | 54.2 / - |
| DRG [13] | ResNet-50 | 51.0 / - | - / - |
| IP-Net [77] | ResNet-50 | 51.0 / - | - / - |
| VSGNet [72] | ResNet-152 | 51.8 / - | 57.0 / - |
| PMFNet [73] | ResNet-50 | 52.0 / - | - / - |
| PD-Net [102] | ResNet-50 | 52.6 / - | - / - |
| CHGNet [74] | ResNet-50 | 52.7 / - | - / - |
| FCMNet [53] | ResNet-50 | 53.1 / - | - / - |
| ACP [34] | ResNet-152 | 53.2 / - | - / - |
| IDN [43] | ResNet-50 | 53.3 / - | 60.3 / - |
| STIP [100] | ResNet-50 | 66.0 / 66.2 | 70.7 / 70.5 |
| *Additional training with object detection data* | | | |
| VCL [24] | ResNet-101 | 48.3 / - | - / - |
| SCG [96] | ResNet-50 | 54.2 / 49.2 | 60.9 / 53.4 |
| UPT [97] | ResNet-101 | 61.3 / 60.3 | 67.1 / 65.6 |
| UniVRD [101] | ViT-L | 65.1 / - | 66.3 / - |
| PViC [98] | Swin-L | 64.1 / - | 70.2 / - |
| RLIPv2 [92] | Swin-L | 72.1 / 71.7 | 74.1 / 73.5 |
| *Single-stage methods* | | | |
| UnionDet [31] | ResNet-50 | 47.5 / - | 56.2 / - |
| HOI-Transformer [108] | ResNet-101 | 52.9 / - | - / - |
| GGNet [103] | Hourglass-104 | 54.7 / - | - / - |
| HOTR [33] | ResNet-50 | 55.2 / 55.0 | 64.4 / 64.1 |
| DIRV [11] | EfficientDet-d3 | 56.1 / - | - / - |
| QPIC [70] | ResNet-101 | 58.3 / - | 60.7 / - |
| CDN [95] | ResNet-101 | 63.9 / 61.3 | 65.8 / 63.2 |
| RLIP [91] | ResNet-50 | 61.9 / 61.3 | 64.2 / 64.0 |
| GEN-VLKT [47] | ResNet-101 | 63.6 / 61.8 | 65.9 / 64.0 |
| ERNet [48] | EfficientNetV2-XL | 64.2 / - | - / - |
| MUREN [35] | ResNet-50 | **68.8** / **68.2** | **71.0** / **70.2** |
| **Ours** | Focal-L | 65.2 / 66.5 | 66.5 / 67.9 |

Table 9: **Quantitative results on V-COCO.** We report both box and mask $mAP$. The best score is highlighted in bold, and the second-best score is underscored. '-' means the model did not release weights and we cannot get the mask $mAP$.

| Model | Trained with original boxes | Trained with SAM masks |
|---|---|---|
| CDN | 31.4 | 28.5 |
| STIP | 32.2 | 29.7 |
| GEN-VLKT | 35.6 | 32.1 |

Table 10: **Results of box $mAP$ on HICO-DET test set.** We train existing HOI detectors with a mask head, by using the masks we generated through SAM.

includes VSGNet, ACP, No-Frills, which use ResNet-152. To the best of our knowledge, larger ResNet, such as ResNet-200, and ResNet-269, are not used in previous methods on related tasks. ResNet, we have included the largest model ResNet-152, which has 65M parameters and 15 GFLOPs. Other baselines are not using the ResNet backbone, for example, UniVRD is using LiT(ViT-H/14) backbone. It has 632M parameters and 162 GFLOPs, a lot more than our 198M parameters and 15.6 GFLOPs, but still performs worse than our model.

## F  Fair comparison of promptable VRS

**Postprocessing of standard VRS outputs.** Since no existing models share the same settings as promptable VRS, we create a baseline for fair comparison. Typically, promptable VRS can be addressed by filtering standard VRS outputs. We post-process outputs from our standard VRS model to extract the desired triplets and compared their $mAP$ with those from promptable VRS. The post-processed results yield a lower $mAP$ (15.7 vs. 26.8), primarily because the selected triplets often have lower confidence scores. Additionally, the post-processing approach is slower, taking 8 seconds compared to 5 seconds for directly prompting the model to retrieve the desired triplet.

**Grounding ability compared with prompt-based vision-language models.** Although promptable VRS is similar to vision-language models like GLIP [40] and MDETR [29] in grounding capabilities, it has distinct objectives. Unlike these models, which focus on entities, promptable VRS outputs triplets, making direct comparisons infeasible. Previous models are not equipped to handle the promptable relationship understanding task directly. To explore this, we modify our structural design to incorporate multiple text prompts as inputs, which are individually processed with their matching scores aggregated for classification. This experimental setup, however, results in reduced performance, increased inference time (26s vs. 5s), and higher GPU memory usage (5G vs. 3G). Thus, we argue that the proposed structure is suitable for tackling promptable VRS.

# G    Masks generated by SAM

**Clarifications of choosing segmentation masks** We firstly illustrate the importance of choosing segmentation masks over boxes in Fig. 7. Traditional bounding boxes often include overlapping and ambiguous information, leading to redundancy. Segmentation masks, by accurately delineating object boundaries, provide a more precise and clear representation, reducing such redundancy, which is also illustrated in [85] and [86]. Besides, segmentation masks provide enhanced visual understanding and comprehensive contextual analysis. Additionally, object detection models often struggle to precisely extract foreground objects, which is why they are typically combined with segmentation models like SAM for fine-grained image tasks. Our model, however, presents a unified model that can localize both subjects and objects, along with their corresponding segmentation masks.

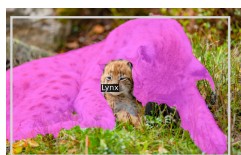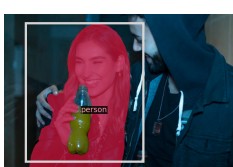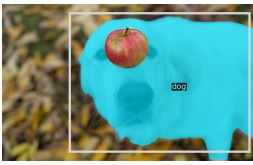

Figure 7: **Illustration of the importance of using masks instead of bounding boxes.** We show examples where one object is occluded by other objects. We show both bounding box annotations and masks generated with SAM, where only the masks can correctly locate the pure object.

**Noise handling in using masks generated by SAM.** To address potential noise and inaccuracies in masks generated by SAM, we employ a filtering approach based on Intersection over Union (IoU). We compute the IoU between the generated masks and the original box annotations. Masks with an IoU score below a threshold of 0.2 are considered to have significant deviations from the ground truth and are filtered out. This threshold is chosen to balance the trade-off between including sufficient mask data and excluding those with substantial inaccuracies. The chosen IoU threshold helps ensure that only masks with a reasonable overlap with the ground truth annotations are retained. This threshold is set based on empirical evaluation and aims to minimize the impact of masks that are too noisy or incorrect, while still retaining as much useful data as possible. After using this strategy, we conduct analysis on 200 samples. We tested various thresholds and found this gets the best balance between denoising and data retaining(95% valid data retraining).

**More visualizations of generated masks.** We have included additional visualizations in Fig. 8 to illustrate the fine-grained masks generated from the bounding box annotations of existing HOI detection datasets. These visualizations indicate that converting to masks significantly reduces the redundancy in the box annotations. Additionally, as shown in Fig. 8 (d), filtering with IoU helps eliminate low-quality masks.

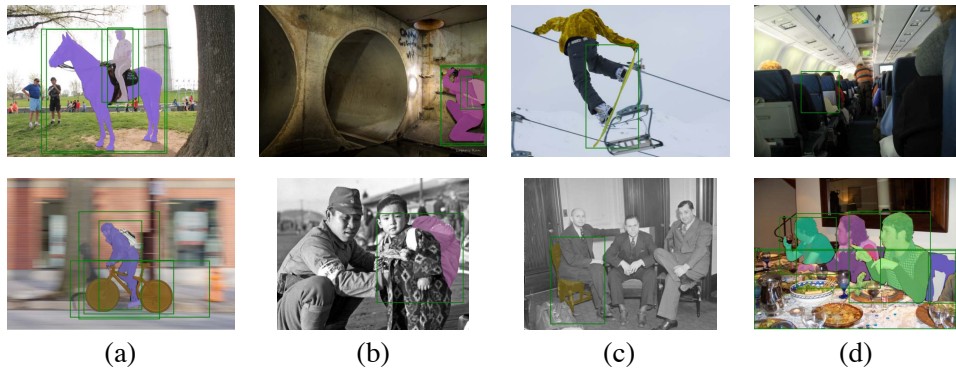

|   (a)   |   (b)   |   (c)   |   (d)   |

Figure 8: **Samples of fine-grained masks generated by converting existing bounding box annotations with SAM.** Samples are chosen from the HICO-DET dataset. Green boxes are original box annotations. Duplicated boxes are suppressed after converting to the mask, as shown in (a). There are also failure cases where no masks are generated with the given box annotations, as in (d).

