# OpenReview forum: "Towards Flexible Visual Relationship Segmentation"
_NeurIPS.cc/2024/Conference — NeurIPS 2024 poster_

### Official Review · Reviewer_k2zH · 2024-07-12

**Soundness:** 3
**Presentation:** 3
**Contribution:** 2
**Rating:** 5
**Confidence:** 4

**Summary:**

This paper proposed a flexible framework that can effectively handle human-object interaction (HOI) detection, scene graph generation (SGG), and referring expression comprehension (REC) tasks. The proposed method further addressed the problem of promptable visual relationship segmentation and enabled the capability for open-vocabulary segmentation. Its main idea is to leverage the pretrained vision-language models for grounding textual features to visual relationship inside images. Experimental results show that the proposed framework is able to achieve SOTA performance on standard, promptable and open-vocabulary visual relationship detection/segmentation tasks.

**Strengths:**

1. This paper is overall well-written and easy to follow.

2. The problem addressed by this paper is novel to me. Unifying various visual relationship detection/segmentation tasks with in a single framework has been seldomly studied in previous works, which might raise new research interests in this field.

3. The overall performance of the method is competitive. It is able to achieve SOTA results on most VRD tasks and outperform previous methods notably.

**Weaknesses:**

1. According to Table 7, the major performance improvement of the proposed method comes from the supervision of the mask head, which is provided from SAM and never used by previous works. When only the box head is used, the results of the proposed method is behind the current SOTA methods notably on HICO-DET and VCOCO. This makes the benefit of the proposed unified framework questionable if such large performance drops are observed.

2. Apart from introducing the segmentation head, the major technical contribution of the proposed method is unclear to me. The model architecture and losses are very similar to those previously proposed dual-decoder architectures in GEN-VLKT, RLIP, etc., and this part needs further clarification.

3. In Table 7, from the last row, we can see that the benefit brought by adding PSG is minor. Hence, the results on PSG dataset should also be reported so that the readers could have a better understanding on how unifying the framework could benefit each individual task.

**Questions:**

Please refer to the Weakness section.

**Limitations:**

The authors have widely discussed the limitation of the proposed work and also its challenges in benchmarking.

---

> ### Author Rebuttal · Authors · 2024-08-06
>
> Thank you for your time and helpful feedback. Please refer to “The General Response to Reviewers” for the reply to the issue of the difference with previous methods and refer to the PDF for the added tables for PSG results and mask head only results. We respond below to your other comments and questions.
>
> 1. **Major performance improvement comes from the mask head.**
>
>     This performance drop can be attributed to the fact that our network architecture, derived from Mask2Former, is optimized for pixel-wise segmentation tasks and does not fully leverage the benefits of box-based supervision.
>     To further investigate, we trained existing HOI detectors (CDN, STIP, GEN-VLKT) with only mask head, using segmentation masks instead of box annotations. Surprisingly, all of these approaches led to reduced accuracy on HICO-DET as in Table 2 of the attached PDF, indicating that their networks’ performance with segmentation masks are not as effective as with box annotations. We argue that our proposed method’s preference for mask annotations is analogous to how previous methods favor box annotations.
>
> 2. **The technical contribution of the proposed method is unclear.**
>
>     While our architecture shares similarities with dual-decoder models like GEN-VLKT and RLIP, our key contributions are:
>
>     (1) ***Unified Framework***: As in Table 1 of the main paper, our model integrates standard, promptable, and open-vocabulary VRS tasks into a single system, offering broader flexibility than previous methods.
>
>     (2) ***Mask-Based Approach***: We utilize masks to handle various VRS tasks, allowing our model to adapt to different types of annotations effectively, including HOI detection and generic scene graph generation.
>
>     (3) ***Dynamic Prompt Handling***: Our approach supports dynamic prompt-based and open-vocabulary settings, addressing limitations of fixed-vocabulary models. As in the Figure 4 of the main paper, our model can even combine the promptable query with open-vocabulary ability to make the model ground novel relational objects.
>
>      Please refer to “The General Response to Reviewers”, where we elaborate more on the difference of our model with previous works.
>
> 3. **Results of PSG in Table 7 of the main paper.**
>
>     Thank you for your suggestion. We have added the results for the PSG dataset in Table 7 of the main paper, please refer to the PDF. The updated results show that while adding HICO-DET and VCOCO can provide slight improvements for PSG, PSG itself does not significantly enhance performance on HICO-DET. This is due to the relatively small size and distinct distributions of these datasets, which may not always be mutually beneficial. We appreciate your input and hope this additional information clarifies the impact of unifying the framework across different tasks.

---

> ### Author Response · Authors · 2024-08-12
>
> Dear Reviewer k2zH,
>
> We sincerely appreciate your recognition and valuable feedback on our paper. In response, we have conducted additional experiments regarding the mask head and the PSG task. We have also provided a detailed clarification of our contributions compared to previous works.
>
> We kindly inquire whether these results and clarifications have adequately addressed your concerns. Please feel free to let us know if you have any further questions.
>
> Thank you a lot!
>
> Best regards,
> Authors of Paper 3507

---

> > ### Comment · Area_Chair_E2Wx · 2024-08-13
> >
> > Dear reviewer, a reminder to take a look at the author's rebuttal and other reviews. Did the rebuttal address your concerns?

---

> > > ### Comment · Reviewer_k2zH · 2024-08-13
> > >
> > > I thank the authors for their responses. However, my major concerns still remain.
> > >
> > > 1. Regarding Response 1, I thank the authors for providing the additional results. These results suggest that the proposed method and previous SOTAs have different focus, but do not address my concerns on whether the major performance improvements of the paper are brought by training models with SAM labelled data. More direct baselines should be training previous SOTA models with the same SAM data as used in the paper, or training the proposed method with the same segementation data used in the mask heads of previous methods.
> > >
> > > 2. Regarding Response 3, from the newly added results, we could observe that the performance gains brought by training with different datasets within a unified framework is very margnial. This make the true benefits of unification unconvincing.
> > >
> > > 3. The authors claim that the proposed method uses much fewer training data (x50 less, without using VG and Objects365) than UniVRD and RLIPv2, but I think this claim does not hold: the proposed model relies on the SAM model which is trained with far more labelled data than VG and Objects365.
> > >
> > > Given the above concerns, I could not recomnad acceptance for the current version of this paper.

---

> > > > ### Author Response · Authors · 2024-08-13
> > > >
> > > > Thanks for the response. We'd like to clarify those concerns.
> > > >
> > > > 1. We want to clarify that the suggested direct baseline — training previous SOTA models with the same SAM data used in our paper — is **exactly** what we provided in the rebuttal PDF, which we have also attached below. It shows that SAM-labeled data can hurt previous models. Thus, the major performance improvements of our work are due to both the SAM-labeled data and our architectural design.
> > > > | Model    | Trained with original boxes | Trained with SAM masks |
> > > > |----------|-----------------------------|------------------------|
> > > > | CDN      |             31.4            |          28.5          |
> > > > | STIP     |             32.2            |          29.7          |
> > > > | GEN-VLKT |             35.6            |          32.1          |
> > > >
> > > >
> > > > 2. We argue that the unified model presented here integrates standard, promptable, and open-vocabulary VRS tasks into a single system, offering broader flexibility than previous methods, without requiring elaborate designs for various tasks. We did not claim that these tasks are mutually beneficial. Thus, we explain why the potential synergies of different datasets are not mutually beneficial in the rebuttal and on Lines 279-285 of the main paper.
> > > >
> > > >
> > > > 3. SAM is treated as an off-the-shelf tool to generate mask annotations, similar to how off-the-shelf detectors like DETR, Faster R-CNN, and YOLO are used to generate box annotations. We did not use SAM training data in our model training. However, previous models incorporate VG and Objects365 into their training processes. These models use box annotations generated by off-the-shelf detectors, but the training data for those detectors is also not counted as part of their models' training data.
> > > >
> > > > If you have any further concerns or questions, please bring it so we can answer it promptly as it is close to the end of the discussion period.
> > > >
> > > > Thanks again,
> > > >
> > > > Authors of Paper 3507

---

> > > > > ### Comment · Reviewer_k2zH · 2024-08-13
> > > > >
> > > > > 1. I thank the authors for the additional clarification. The results in the current setup make more sense and convincing, which addressed my major concerns on the performance side.
> > > > >
> > > > > 2. I appreciate the authors' efforts on designing a unified framework for VRS tasks. Ideally, one of the major benefits/motivations of unification is that a unified framework usually enables training with more data in a flexible way and lead to performance improvement across different benchmarks. I suggest the authors to have a more extensive discussion on this perspective in the limitation and future work sections of the revised paper.
> > > > >
> > > > > 3. I have to point out that the detectors (e.g., DETR, Faster R-CNN, and YOLO) used by previous work are usually trained on small scale datasets like COCO, while SAM is trained with far larger-scale data than even the combination of COCO, VG and Objects365. Therefore, the two configurations here are not directly comparable.
> > > > >
> > > > > Since my major concerns have been addressed, I would like to increase my rating to 5.

---

> > > > > > ### Author Response · Authors · 2024-08-14
> > > > > >
> > > > > > Dear Reviewer k2zH,
> > > > > >
> > > > > > We appreciate your prompt feedback. Thank you for your suggestion; we will definitely include a discussion on the mutual benefits of the unified model in the limitations and future work section of our revised version. We agree that SAM has been trained on significantly larger-scale data compared to the training datasets of the detectors.
> > > > > >
> > > > > > However, we would like to clarify that in our comparison, we focus on the number of images used directly to train our model, not the training data used to obtain the model(SAM) for pseudo-labeling. Additionally, we directly utilize the SAM model to generate pseudo-mask labels, whereas other methods rely on both off-the-shelf detectors and human labor for box labeling as the teacher model. We hope this helps clarify the third point further.
> > > > > >
> > > > > > Thanks again for your time and dedication!
> > > > > >
> > > > > > Authors of Paper 3507

---

### Official Review · Reviewer_EDLE · 2024-07-14

**Soundness:** 3
**Presentation:** 3
**Contribution:** 3
**Rating:** 7
**Confidence:** 4

**Summary:**

This work presents a novel approach for visual relationship segmentation that integrates the three critical aspects of a flexible VRS model: standard VRS, promptable querying, and open-vocabulary capabilities. By harnessing the synergistic potential of textual and visual features, the proposed model delivers promising experimental results on existing benchmark datasets.

**Strengths:**

1) The authors introduce a flexible framework capable of segmenting both human-centric and generic visual relationships across various datasets.
2) The authors present a promptable visual relationship learning framework that effectively utilizes diverse textual prompts to ground relationships.
3) The proposed method shows competitive performance in both standard close-set and open-vocabulary scenarios, showcasing the model’s strong generalization capabilities.

**Weaknesses:**

1. The experimental comparison suffers from potential unfairness, contrasting the proposed method with others using Focal-L backbone against those employing ResNet backbone seems somewhat unreasonable.
2. Since this paper includes referring expression comprehension (REC) tasks in abstract, it is unreasonable not to report experimental results on corresponding benchmarks like RefCOCO, RefCOCO+, and RefCOCOg.
3. While the research perspective of this paper is reasonable and innovative, the overall architecture design of Flex-VRS lacks novelty compared to previous related works.
4. More visualizations are needed for demonstrating the produced fine-grained masks generated by converting existing bounding box annotations from HOI detection datasets.

**Questions:**

Please refer to Weakness Section for more details.

**Limitations:**

The authors have thoroughly discussed the limitations of this work.

---

> ### Author Rebuttal · Authors · 2024-08-07
>
> Thank you for your time and helpful feedback. Please refer to “The General Response to Reviewers” for the reply to the issue of backbone comparison, difference with previous methods and refer to the PDF for the added visualizations of masks generated from bounding box annotations. We respond below to your other comments and questions.
>
> 1. **Results on referring expression comprehension (REC) task.**
>     - The referring expression comprehension (REC) tasks on benchmarks like RefCOCO, RefCOCO+, and RefCOCOg are designed to detect objects based on free-form textual phrases, such as "a ball and a cat" or "Two pandas lie on a climber." In contrast, the promptable VRS task in our work focuses on detecting subject-object pairs within a structured prompt format, such as <?, sit_on, bench> or <person, ?, horse>, as illustrated in Figure 1 of the main paper. Our model is designed to encode and compute similarity scores for each of these elements separately.
>     - Adapting our model to standard REC tasks would require fundamental changes, as the REC format does not align with the structured, relational queries that our model is designed to handle. Our primary focus is on relational object segmentation based on a single structured query, which differs significantly from the objectives of REC benchmarks. Therefore, it is not reasonable to directly compare our results with those benchmarks without substantial modifications to the model.
>
> 2. **The architecture design lacks novelty.**
>
>     - As highlighted in Table 1 of the main paper, our Flex-VRS framework introduces a flexible architecture that supports standard, promptable, and open-vocabulary visual relationship learning. The novelty of our approach lies in the seamless integration of these diverse functionalities within a single, unified system. This design enables flexible and dynamic interactions with visual relationships, setting our work apart from previous methods that often lack such comprehensive capabilities.
>
>     - Furthermore, our model's one-stage design allows it to effectively handle different types of inputs and perform various tasks based on the input, thanks to our novel query mechanism. This innovative query design is a significant departure from previous works, offering a level of adaptability and flexibility that has not been previously developed.
>
> 3. **More visualizations of generating masks from bounding box annotations.**
>
>     Thank you for the suggestion. We've included additional visualizations in the attached PDF. These visualizations demonstrate how redundant bounding boxes are consolidated into a single mask, and how our IoU-based filtering effectively removes failure cases.

---

> ### Author Response · Authors · 2024-08-13
>
> Dear Reviewer EDLE,
>
> We sincerely appreciate your valuable feedback and suggestions. In our rebuttal, we have provided explanations regarding the experimental comparisons, differences from REC tasks and previous works, and have included additional visualizations.
>
> We kindly inquire whether these results and clarifications have adequately addressed your concerns. Please feel free to let us know if you have any further questions.
>
> Thank you very much!
>
> Best regards,
> Authors of Paper 3507

---

> > ### Comment · Area_Chair_E2Wx · 2024-08-13
> >
> > Dear reviewer, a reminder to take a look at the author's rebuttal and other reviews. Did the rebuttal address your concerns?

---

> > > ### Author Response · Authors · 2024-08-14
> > >
> > > Dear Reviewer EDLE,
> > >
> > > Thank you again for your time to review our paper. Could you please check if our rebuttal has addressed your concerns at your earliest convenience? The deadline of the discussion period will end very soon. Thank you!
> > >
> > > Best regards,
> > >
> > > Authors of Paper 3507

---

> ### Comment · Reviewer_EDLE · 2024-08-14
> **Response to Authors**
>
> The authors have addressed most of my concents. After checking the peer review comments and the author's responses, I decided to raise the given score for this work.

---

### Official Review · Reviewer_W5TQ · 2024-07-15

**Soundness:** 3
**Presentation:** 3
**Contribution:** 3
**Rating:** 5
**Confidence:** 5

**Summary:**

This work proposes an approach for visual relationship segmentation that integrates the three aspects of a VRS model: standard VRS, promptable querying, and open-vocabulary capabilities. The idea of the article is very good, but the performance seems to be lacking.

**Strengths:**

Enhancing HOI from the perspective of object segmentation is an interesting and promising idea.

**Weaknesses:**

1-Previous work mostly uses ResNet series backbones, while the authors use a Focal-L Backbone. How do the FLOPs and parameter counts of this backbone compare to ResNet? I am concerned there may be an unfair comparison.

2-The performance of the proposed method does not seem to be particularly superior, which is a significant drawback. For example, in Tables 2 and 3, UniVRD, PViC, and RLIPv2 significantly outperform the proposed method in terms of box mAP metrics. Can the authors analyze this situation?

3-In line 208, the authors mention using SAM to generate masks. How do they handle the noise in these masks? Some masks might significantly deviate from the ground truth.

4-There are some typos, such as in line 25: "textttperson".

**Questions:**

see weakness

**Limitations:**

see weakness

---

> ### Author Rebuttal · Authors · 2024-08-06
>
> Thank you for your time and helpful feedback. Please refer to “The General Response to Reviewers” for the reply to comparison of our backbone with ResNet's and refer to the PDF for visualizations of masks generated by SAM. We respond below to your other comments and questions.
>
> 1. **Performance seems not to be particularly superior. Can the authors analyze this situation?**
>
>
>     (1) ***Training Data and Model Efficiency***: Our approach is designed to be more data-efficient and requires fewer training data compared to methods like RLIPv2 and UniVRD, which use extensive datasets. For example, they used both VG (108,077 images) and Objects365 (more than 2,000,000 images), while our model only used single data source, e.g. HICO-DET (47,776 images), which is x50 smaller. The fact that our model does not rely on these large datasets impacts the box mAP performance in direct comparisons.
>
>     (2) ***Model Complexity and Two-Stage Approach***: UniVRD, PViC, and RLIPv2 employ two-stage methods and fine-tune object detectors using extra datasets, which contribute to their higher box mAP scores. In contrast, our model uses a one-stage approach without separate object detection fine-tuning. This simple design choice, while impacting box mAP, aligns with our goal of creating a more flexible and data-efficient model.
>
>     (3) ***Model Capacity***: Our model, utilizing the Focal-L backbone, is significantly smaller in terms of capacity compared to UniVRD. For instance, the Focal-L backbone has 198M parameters compared to 632M in UniVRD with the LiT (ViT-H/14) model. Despite this difference, we achieved better results in terms of overall performance metrics, such as 37.4 vs 38.1 on HICO-DET, which highlights the efficiency and effectiveness of our approach despite its smaller size.
>
>     To conclude, our model prioritizes data efficiency and simplicity over sheer complexity, utilizing fewer training data and a one-stage approach without additional fine-tuning on extra datasets. Despite having a significantly smaller model capacity, our approach still achieves comparable overall performance, though at the cost of slightly lower box mAP scores. This trade-off reflects our focus on creating a more flexible and efficient model rather than optimizing for specific metrics like box mAP.
>
>
> 2. **Noise handling in using masks generated by SAM.**
>
>     To address potential noise and inaccuracies in masks generated by SAM, we employ a filtering approach based on Intersection over Union (IoU).
>     - IoU-Based Filtering: We compute the IoU between the generated masks and the original box annotations. Masks with an IoU score below a threshold of 0.2 are considered to have significant deviations from the ground truth and are filtered out. This threshold is chosen to balance the trade-off between including sufficient mask data and excluding those with substantial inaccuracies.
>     - Rationale for IoU Threshold: The chosen IoU threshold helps ensure that only masks with a reasonable overlap with the ground truth annotations are retained. This threshold is set based on empirical evaluation and aims to minimize the impact of masks that are too noisy or incorrect, while still retaining as much useful data as possible. Please refer to the attached PDF for visualizations of generated masks from bounding box annotations. After using this strategy, we conduct analysis on 200 samples. We tested various thresholds and found this gets the best balance between denoising and data retaining(95% valid data retraining).
>
>
> 3. **Typos.**
>
>     Thank you for pointing out the typo. We will ensure that the final version of the manuscript is thoroughly proofread to address any such errors.

---

> ### Author Response · Authors · 2024-08-12
>
> Dear Reviewer W5TQ,
>
> We sincerely appreciate your recognition and valuable feedback on our paper. In response, we have analyzed the backbone issue and provided detailed comparisons and evaluation clarifications of our model against previous works. Additionally, we have clarified the handling of noise in masks and will ensure this is clearly outlined in our revised version.
>
> We kindly inquire whether these clarifications have adequately addressed your concerns. Please feel free to let us know if you have any further questions.
>
> Thank you a lot!
>
> Best regards,
> Authors of Paper 3507

---

> > ### Comment · Area_Chair_E2Wx · 2024-08-13
> >
> > Dear reviewer, a reminder to take a look at the author's rebuttal and other reviews. Did the rebuttal address your concerns?

---

> > > ### Comment · Reviewer_W5TQ · 2024-08-14
> > >
> > > The author has addressed my concerns, and I will increase the rating to 5

---

> > > > ### Author Response · Authors · 2024-08-14
> > > >
> > > > Dear Reviewer W5TQ,
> > > >
> > > > We sincerely appreciate your thorough and professional review of our paper. We're pleased that our rebuttal addressed your concerns, and we're grateful for your recognition and timely response.
> > > >
> > > > Best regards,
> > > >
> > > > Authors of Paper 3507

---

### Official Review · Reviewer_LLLd · 2024-07-16

**Soundness:** 2
**Presentation:** 2
**Contribution:** 2
**Rating:** 5
**Confidence:** 4

**Summary:**

This paper propose a model to handle multiple visual relationship tasks, like HOI detection and Scene Graph Generation.

The proposed model is based on vision-language models similar to CLIP.

It handles different formulations like standard close-set, open-vocabulary, and prompted setting.

**Strengths:**

- Unified model for HOI and SGG.
- Multiple inference setting supported, like open-vocabulary, standard close-set, and prompted.
- The writing is smooth, and the demonstration is generally OK.

**Weaknesses:**

- Why not detection, rather than segmentation? What's the necessity for segmentation-style output rather than traditional detection style tasks? I would like more clarification on this.
- Also, (if not using the segmentation setting,) this method could be compared with more methods. Currently, the compared methods seems still not sufficient.
- This general contribution and pipeline is similar to UniVRD, except (1) this model is mask-based rather than box-based, and (2) a prompted setting is further supported. Thus, the contribution seems incremental.
- The performance is basically the same as UniVRD and is not as good as RLIPv2.

**Questions:**

- What's the essential difference between this model and UniVRD?
- What's the necessity for segmentation rather than prediction?

**Limitations:**

Limitations are discussed.

---

> ### Author Rebuttal · Authors · 2024-08-06
>
> Thank you for your time and helpful feedback. Please refer to “The General Response to Reviewers” for the reply to the issue of the difference with previous methods and refer to the PDF for illustration of the importance of using segmentation masks. We respond below to your other comments and questions.
>
> 1. **Advantages of segmentation over detection.**
>
>    - Precision and Reduced Redundancy: Traditional bounding boxes often include overlapping and ambiguous information, leading to redundancy. Segmentation masks, by accurately delineating object boundaries, provide a more precise and clear representation, reducing such redundancy, which is also illustrated in [1] and [2].
>     - Enhanced Visual Understanding: Segmentation allows for a more detailed analysis of visual relationships, crucial for tasks like panoptic scene graph generation and human-object interaction detection. It enables the model to better distinguish between object parts and backgrounds, leading to improved performance in complex scene understanding. As in Table 7 of the main paper, using box head leads to performance drop in 7.5 mAP on HICO-DET and 4.5 mAP on VCOCO.
>     - Comprehensive Contextual Analysis: Bounding boxes typically miss important contextual elements, such as background categories like roads, sky, and trees, as shown in the standard PSG task in Figure 1 of the main paper.  Segmentation captures these elements, offering a more complete understanding of the scene, which is vital for tasks like image and video editing.
>     - Unified Model: Additionally, object detection models often struggle to precisely extract foreground objects, which is why they are typically combined with segmentation models like SAM for fine-grained image tasks. Our approach, however, presents a unified model that can localize both subjects and objects, along with their corresponding segmentation masks. We show visualization to illustrate this point in the attached PDF.
>
>      We hope this clarifies the advantages of using segmentation masks in our approach.
>
> 2. **Could Compare with more methods if not using the segmentation setting.**
>
>     Thank you for the valuable feedback! We actually conducted comparisons without the segmentation setting, as detailed in the paper. Besides computing mask mAP in segmentation setting, we also conducted box mAP in non-segmentation setting. We did this by converting our segmentation masks into bounding boxes and transforming the box outputs of previous methods into masks. In Table 2 and 3 in the main paper, we show comparison with 13 previous methods, including one-stage methods, two-stage methods, with different backbones(ResNet, Swin, ViT) and training data, and in Table 8 and 9 in the appendix, we show full comparison with over 25 methods.
>
> [1] Yang, Jingkang, et al. "Panoptic scene graph generation." ECCV 2022.
>
> [2] Yang, Jingkang, et al. "Panoptic video scene graph generation." CVPR 2023.

---

> ### Author Response · Authors · 2024-08-13
>
> Dear Reviewer LLLd,
>
> We sincerely appreciate your valuable feedback on our paper. In our rebuttal, we have discussed the differences between our work and previous studies, as well as provided clarifications regarding the segmentation of relational objects.
>
> We kindly inquire whether these clarifications have adequately addressed your concerns. Please feel free to let us know if you have any further questions.
>
> Thank you very much!
>
> Best regards,
> Authors of Paper 3507

---

> > ### Comment · Area_Chair_E2Wx · 2024-08-13
> >
> > Dear reviewer, a reminder to take a look at the author's rebuttal and other reviews. Did the rebuttal address your concerns?

---

> > > ### Author Response · Authors · 2024-08-14
> > >
> > > Dear Reviewer LLLd,
> > >
> > > Thank you again for your time to review our paper. Could you please check if our rebuttal has addressed your concerns at your earliest convenience? The deadline of the discussion period will end very soon. Thank you!
> > >
> > > Best regards,
> > >
> > > Authors of Paper 3507

---

### Author Rebuttal · Authors · 2024-08-06

We thank the reviewers for their thoughtful feedback. Our proposed Flex-VRS is commended for its ability to support a variety of tasks(Reviewer LLLd, EDLE, k2zH), its integration of flexibility and open-vocabulary capability into the VRS model(Reviewer LLLd, EDLE), its competitive performance(Reviewer EDLE, k2zH), and its clear presentation(Reviewer LLLd, k2zH). We address some common questions from reviewers here and will incorporate the feedback in the revision. Please also refer to the attached PDF for added visualizations of generated masks and tables for PSG results and mask head results.

1. **Difference from previous methods (e.g. UniVRD).**

    We would like to clarify the contributions and differences between our method and previous work, including UniVRD and RLIPv2.

    - **Comparison with UniVRD** – *Methodology Difference*: UniVRD uses a two-stage approach, where the model first detects independent objects and then decodes relationships between them, retrieving boxes from the initial detection stage. In contrast, our method employs a one-stage approach, where each query directly corresponds to a <subject, object, predicate> triplet. This transition improves time efficiency from O(MxN) to O(K), where M is the number of subject boxes, N is the number of object boxes, and K is the number of interactive pairs. ***Our approach also provides greater flexibility by learning a unified representation that encompasses object detection, subject-object association, and relationship classification in a single model.*** (As shown in Figure 1, the subject-object pairs need to be accurately localized and understood.) Our model is superior in terms of model capacity and data efficiency. While we use much fewer training data (x50 less, without using VG and Objects365) and our model with the Focal-L backbone is much smaller than UniVRD (164M vs 640M) with LiT(ViT-H/14), we achieve comparable results(37.4 vs 38.1 on HICO-DET).

     - **Performance Comparison with RLIPv2** – *Scaling and Design*: While our method does not match RLIPv2 in performance, this is due to different design philosophies and goals. RLIPv2 is a two-stage approach optimized for large-scale pretraining and relies on separately trained detectors. ***Our model, however, is not designed for pretraining and does not include a separately trained detector. Our focus is on enhancing the flexibility of the VRS model without relying on extensive curated data(x50 more, VG and Objects365).*** Thus, the differences in performance are attributed to the scale and design objectives rather than a direct comparison of methods.

2. **FLOPs and the number parameters of the backbone compared to ResNet.**

    As in Table 2 and 3 of the main paper, we have done extensive comparisons with previous methods, including backbones on ResNet-50/101/152, EfficientNet, Hourglass, Swin Transformers, and LiT architectures. For previous methods that utilize ResNet backbone for HOI detection and PSG, we already conducted a thorough comparison and included (VSGNet, ACP, No-Frills), which used ResNet-152.  To the best of our knowledge,  larger ResNet, such as ResNet-200, and ResNet-269, are not used in previous methods on related tasks. ResNet, we have included the largest model ResNet-152, which has 65M parameters and 15 GFLOPs.

    Other baselines are not using the ResNet backbone, for example, UniVRD is using LiT(ViT-H/14) backbone. It has 632M parameters and 162 GFLOPs, a lot more than our198M parameters and 15.6 GFLOPs,  but still performs worse than our model.

---

### Author Response · Authors · 2024-08-14
**Summary of the author-reviewer discussion**

We would like to express our sincere thanks to all the reviewers for their detailed and insightful feedback and suggestions. We also extend our gratitude to the ACs for organizing the review process for submission 3507. After rebuttal, we receive positive ratings from all reviewers.

We propose a flexible model that seamlessly integrates standard, promptable, and open-vocabulary visual relationship segmentation, delivering promising experimental results and demonstrating a high degree of adaptability. We would also like to highlight how we address the reviewers' concerns at the conclusion of the discussion period.

For reviewer LLLd, we resolve the concerns by: 1) clarifying the advantages of segmentation over detection; 2) clarifying how we compare with previous works under different settings; and 3) discussing the differences with previous works.

For reviewer W5TQ, we address the concerns by: 1) providing backbone comparisons; 2) discussing performance comparisons; and 3) elaborating on the noise handling of generated masks. Reviewer W5TQ acknowledged that our rebuttal addressed their concerns and subsequently raised the score.

For reviewer EDLE, we tackle the concerns by: 1) providing backbone comparisons; 2) clarifying our architectural design; 3) discussing the differences between our task and REC tasks, and we will change the term in the revised version to avoid confusion; and 4) showing additional visualizations of the generated masks.

For reviewer k2zH, we address the concerns by: 1) providing additional results for baseline comparisons; 2) providing further results on PSG along with clarifications; and 3) discussing the training data volume of our model compared to previous methods. Reviewer k2zH acknowledged that our rebuttal addressed the major concerns and subsequently raised the score.

Overall, our model integrates various functionalities into a single framework, and its effectiveness is validated under different settings, including standard, promptable, and open-vocabulary VRS, as well as using different supervisions, losses, and datasets for fair comparisons.

We will incorporate all suggestions from the reviewers into our revised version.

Best regards,

Authors of Paper 3507

---

### Decision · Program_Chairs · 2024-09-25

**Decision:**

Accept (poster)

**Comment:**

This paper was reviewed by four experts in the field. The authors did a great job in the rebuttal, and reviewers raised their ratings after the rebuttal (unanimously agreed to accept the paper). The reviewers did raise some valuable suggestions in the discussion that should be incorporated in the final camera-ready version of the paper. The authors are encouraged to make the necessary changes to the best of their ability.

Why not higher: Reviewer k2zH raised some valuable concerns, including more discussions on the unified framework for VRS tasks and using SAM-based models to compare with other models trained on much smaller datasets.